# Tribbles1 is host protective during in vivo mycobacterial infection

**Ffion R Hammond[1], Amy Lewis[1], Gabriele Pollara[2], Gillian S Tomlinson[2], Mahdad Noursadeghi[2], Endre Kiss-Toth[1]\*, Philip M Elks[1]\***

[1]The Bateson Centre, School of Medicine and Population Health, Faculty of Health, University of Sheffield, Sheffield, United Kingdom; [2]Division of Infection & Immunity, University College London, London, United Kingdom

**Abstract** Tuberculosis is a major global health problem and is one of the top 10 causes of death worldwide. There is a pressing need for new treatments that circumvent emerging antibiotic resistance. *Mycobacterium tuberculosis* parasitises macrophages, reprogramming them to establish a niche in which to proliferate, therefore macrophage manipulation is a potential host-directed therapy if druggable molecular targets could be identified. The pseudokinase Tribbles1 (Trib1) regulates multiple innate immune processes and inflammatory profiles making it a potential drug target in infections. Trib1 controls macrophage function, cytokine production, and macrophage polarisation. Despite wide-ranging effects on leukocyte biology, data exploring the roles of Tribbles in infection in vivo are limited. Here, we identify that human Tribbles1 is expressed in monocytes and is upregulated at the transcript level after stimulation with mycobacterial antigen. To investigate the mechanistic roles of Tribbles in the host response to mycobacteria in vivo, we used a zebrafish *Mycobacterium marinum* (Mm) infection tuberculosis model. Zebrafish Tribbles family members were characterised and shown to have substantial mRNA and protein sequence homology to their human orthologues. *trib1* overexpression was host-protective against Mm infection, reducing burden by approximately 50%. Conversely, *trib1* knockdown/knockout exhibited increased infection. Mechanistically, *trib1* overexpression significantly increased the levels of proinflammatory factors *il-1β* and nitric oxide. The host-protective effect of *trib1* was found to be dependent on the E3 ubiquitin kinase Cop1. These findings highlight the importance of Trib1 and Cop1 as immune regulators during infection in vivo and suggest that enhancing macrophage TRIB1 levels may provide a tractable therapeutic intervention to improve bacterial infection outcomes in tuberculosis.

## Editor's evaluation

This is a valuable study that defines the role of the pseudokinase Tribbles1 in the host response to mycobacteria. The data supports the function of Tribbles in a host protective, Cop-1 dependent, inflammatory response to Mycobacterium marinum in zebrafish and is deemed solid. This study would be of interest to scientists focused on host-pathogen interactions.

## Introduction

With the rise of antimicrobial resistance (AMR), bacterial infections are a major threat to global public health. Tuberculosis, caused by the human pathogen *Mycobacterium tuberculosis*, is a case in point, with 1.6 million deaths worldwide (***WHO, 2022***), many of which are resistant to first- and second-line antibiotic treatments (***Allué-Guardia et al., 2021***; ***Hameed et al., 2018***; ***Migliori et al., 2013***). To successfully combat AMR there is a pressing and urgent need for alternative treatment strategies to failing antimicrobials. One such approach is offered by the development of host-derived therapies

**\*For correspondence:**
e.kiss-toth@sheffield.ac.uk (EK-T);
p.elks@sheffield.ac.uk (PME)

**Competing interest:** The authors declare that no competing interests exist.

(HDTs), which target systems in the host rather than the pathogen, circumventing AMR (*Kaufmann et al., 2018*; *Kilinç et al., 2021*).

One primary immune defence against *Mycobacterium tuberculosis* (*Mtb*) is macrophages. Macrophages have a spectrum of phenotypes ranging from proinflammatory to anti-inflammatory, determined in a process known as macrophage polarisation. *Mtb* is expert at manipulation of macrophage polarisation to its advantage (*Ahmad et al., 2022*) and can inhibit the polarisation of proinflammatory macrophages, subverting killing mechanisms to promote intracellular survival of the bacteria and subsequent granuloma formation (*Hackett et al., 2020*). Reprogramming macrophages to better kill *Mtb* is a potential HDT strategy that may be particularly effective against intracellular pathogens (*Sheedy and Divangahi, 2021*).

Tribbles genes encode for a family of pseudokinases (TRIB1, TRIB2, and TRIB3; *Kiss-Toth et al., 2004*), involved in the regulation of core cellular processes, ranging from cell cycle to glucose metabolism (*Grosshans and Wieschaus, 2000*; *Mata et al., 2000*; *Seher and Leptin, 2000*). The TRIB1 isoform has been strongly associated with macrophage roles in inflammation and innate immunity (*Johnston et al., 2015*; *Niespolo et al., 2020*). TRIB1 regulates multiple important macrophage regulatory factors, especially controlling the proinflammatory response, such as tumour necrosis factor alpha (TNF-α), interleukin-1beta (IL-1β), and nitric oxide (NO) (*Arndt et al., 2018*; *Liu et al., 2013*). *Trib1*$^{-/-}$ mice have decreased expression levels of inflammation-related genes such as *IL-6*, *IL-1b*, and *Nos2* (encodes for inducible nitric oxide synthase, iNOS), and murine *Trib1*$^{-/-}$ bone marrow-derived macrophages have defective inflammatory, phagocytic, migratory, and NO responses in vitro (*Arndt et al., 2018*; *Liu et al., 2013*).

TRIB1 influences inflammatory and immune processes via multiple mechanisms. The best described is via recruitment and binding of the E3 ubiquitin ligase constitutive photomorphogenic 1 (COP1). The TRIB1 protein possesses two functional binding sites in its C-terminal, one for COP1 and the second for mitogen-activated protein kinase kinase (MEK) binding. TRIB1 can act as a protein scaffold, binding a substrate to its pseudokinase domain, as well as binding in the functional C terminus to create a regulatory complex. Binding of TRIB1 to the E3 ubiquitin ligase COP1 causes a conformational change, enhancing COP1 binding and bringing COP1 into proximity with the substrate allowing ubiquitination and subsequent degradation (*Jamieson et al., 2018*; *Kung and Jura, 2019*; *Murphy et al., 2015*; *Zahid et al., 2022*). The TRIB1/COP1 complex is responsible for the regulation of multiple targets such as transcription factors, including the tumour suppressor CCAAT/enhancer-binding protein (C/EBPα), which regulates macrophage migration and TNF-α production (*Liu et al., 2013*; *Yoshida et al., 2013*).

While TRIB1 has been shown to regulate several inflammatory and innate immune functions in vitro, its role in infection is much less characterised, especially in an in vivo setting. *TRIB1* is a predicted target of microRNA–gene interactions that differentiate active and latent TB patients (*Wu et al., 2014*) and is an overabundant transcript in highly proinflammatory tuberculosis-immune reconstitution inflammatory syndrome patients (*Lai et al., 2015*). However, despite these reported potential links between Tribbles and TB, interrogation of *TRIB* isoform transcripts in human mycobacterial datasets had not been performed.

Over the last two decades, the zebrafish has proved a powerful model for understanding host–pathogen interactions, due to its high-fecundity, transparency of larvae and availability of transgenic reporter lines. A human disease-relevant and tractable infection model is the zebrafish model of tuberculosis, utilising the injection of the natural fish pathogen *Mycobacterium marinum* (Mm), a close genetic relative of *Mtb* (*Davis et al., 2002*; *van der Sar et al., 2009*). This model has shed light on numerous immune pathways involved in host defence, for example hypoxia-inducible factor (HIF) signalling (*Elks et al., 2013*; *Ogryzko et al., 2019*; *Schild et al., 2020*).

Here, we show that Tribbles1 is expressed in human primary monocytes and its expression is increased at the site of a human in vivo mycobacterial antigen challenge, indicative of a role in TB responses. To substantiate the importance of *TRIB1* in TB pathogenesis, we report a new, protective role for *trib1* in infection defence using an in vivo zebrafish *Mycobacterium marinum* (Mm) infection model. Overexpression of *trib1* significantly reduced Mm burden and increased production of the proinflammatory cytokine *il1b* and NO. The antimicrobial effect of *trib1* overexpression was found to be dependent on *cop1*. Our findings uncover a role for *trib1* in mycobacterial infection defence in vivo, highlighting Trib1 as a potential therapeutic target for manipulation to improve bacterial infection outcomes.

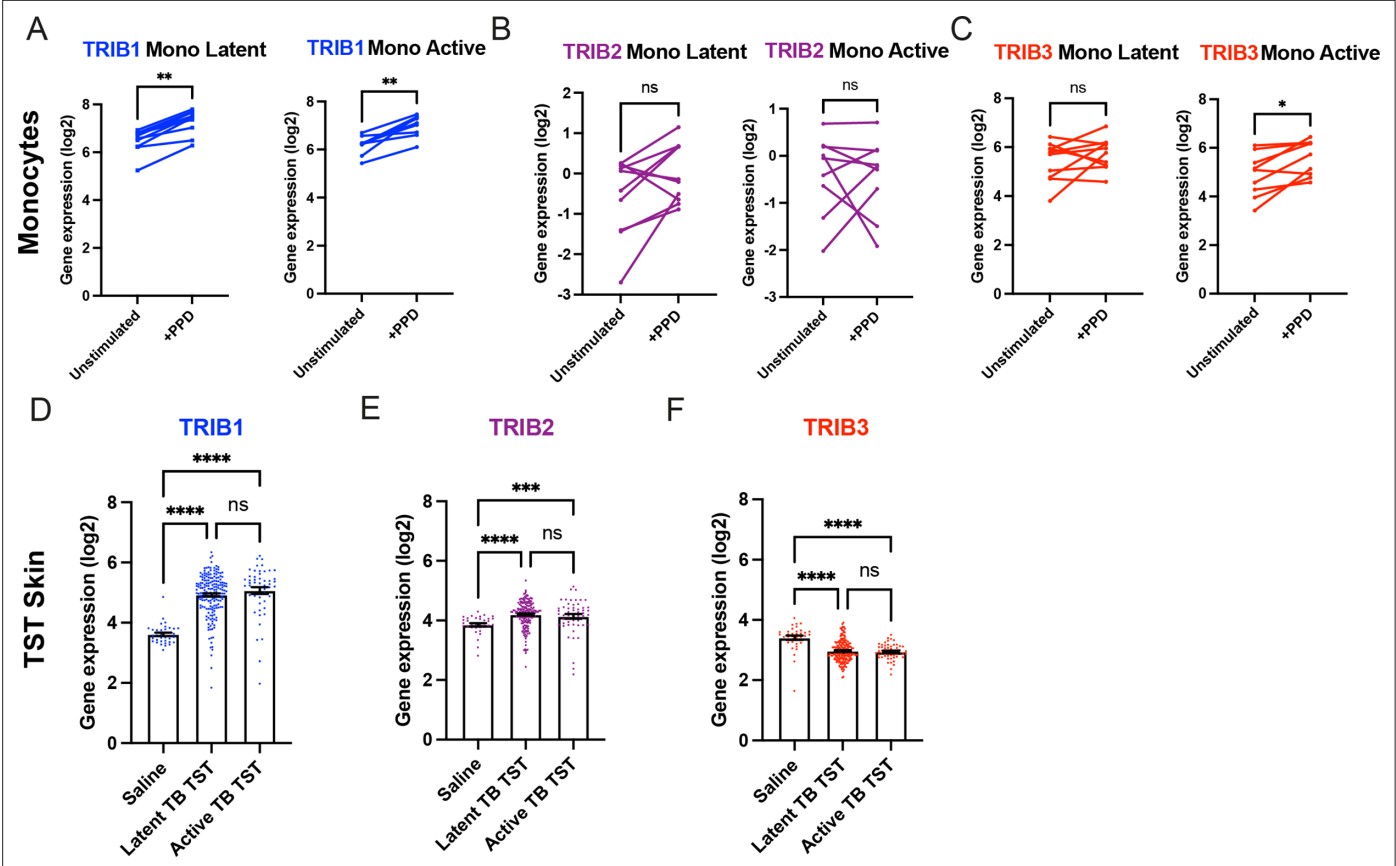

**Figure 1.** Expression of *TRIB1* in human monocytes and tissues is elevated after mycobacterial antigen stimulation. (**A–C**) Expression of *TRIB1*, *TRIB2*, and *TRIB3* transcripts in human CD14$^+$ monocytes in patients with active or latent TB before and after 4 hr of *Mtb* protein derivative (PPD) stimulation in vitro. Each paired data point represents one individual, with active or latent TB (*n* = 9 and *n* = 7, respectively). Statistical significance determined by paired Wilcoxon tests. p values shown are: *p < 0.05 and **p < 0.01. (**D–F**) Expression of *TRIB1*, *TRIB2*, and *TRIB3* within in saline injected human skin and from biopsies of the site of a tuberculin skin test (TST) in patients with active or latent TB. Each point represents one individual with bars for each group representing mean gene expression. *n* = 48 and 191 individuals with active or latent TB, respectively. Statistical significance determined via Kruskal–Wallis with multiple comparisons. p values shown are: ***p < 0.001 and ****p < 0.0001.

The online version of this article includes the following source data for figure 1:

**Source data 1.** Numerical data for the graph of *Figure 1A*, left panel.

**Source data 2.** Numerical data for the graph of *Figure 1A*, right panel.

**Source data 3.** Numerical data for the graph of *Figure 1B*, left panel.

**Source data 4.** Numerical data for the graph of *Figure 1B*, right panel.

**Source data 5.** Numerical data for the graph of *Figure 1C*, left panel.

**Source data 6.** Numerical data for the graph of *Figure 1C*, right panel.

**Source data 7.** Numerical data for the graph of *Figure 1D*.

**Source data 8.** Numerical data for the graph of *Figure 1E*.

**Source data 9.** Numerical data for the graph of *Figure 1F*.

## Results

### *TRIB1* is expressed in human monocytes and is upregulated after in vivo mycobacterial antigen stimulation

To explore whether Tribble pseudokinase expression is modulated by mycobacterial antigen exposure in humans, we initially focused on CD14$^+$ monocytes stimulated in vitro with *Mtb* protein derivative (PPD). This revealed that mycobacterial antigen exposure induced the expression of TRIB1 isoform transcripts but not TRIB2, which had the lowest baseline expression, nor TRIB3, observations

consistent across monocytes from either active or latent TB individuals (*Figure 1A–C*). To determine whether Tribbles play a role in human responses in vivo, we turned to the transcriptomic profiles of biopsies from the site of a tuberculin skin test (TST), a routine clinical investigation repurposed into a mycobacterial antigen challenge model (*Bell et al., 2016*). This revealed baseline expression of TRIB1 in control saline injected tissue samples (*Figure 1D*). Exposure to tuberculin-induced robust induction of TRIB1 expression in TST reactions for both active and latent TB individuals (*Figure 1D*). A more modest increase was seen for TRIB2 (*Figure 1E*) but not for TRIB3 (*Figure 1F*; *Pollara et al., 2021*).

Together these data reveal that TRIB1, and to a lesser extent TRIB2, expression is increased in response to both in vitro and in vivo mycobacterial antigen exposure in humans, independent of clinical TB disease grouping. We interpret these data as signifying a potential functional role for these pseudokinase in regulation of mycobacterial infections in vivo, and indicating the need for a tractable in vivo model of mycobacterial infection to study this further.

## Zebrafish Tribbles isoforms share homology with their human and mouse counterparts and are expressed in immune cell populations

To explore the functional role in vivo for Tribbles in the control of mycobacterial infections in vivo, we developed a zebrafish model of *M. marinum* infection and tools to manipulate *tribbles* expression.

Zebrafish have a single orthologue of each mammalian tribbles isoform, with *tribbles1* (ENSDARG00000110963/previously ENSDARG00000076142), *tribbles2* (ENSDARG00000068179), and *tribbles3* (ENSDARG00000016200) genes. The exon organisation of Tribbles genes is conserved between human and zebrafish *trib* isoforms. Zebrafish *trib1* has three exons like murine *Trib1* and human *TRIB1* (*Figure 2A*). Human *TRIB2* and mouse *Trib2* also share this exon structure but are larger than the TRIB1 isoforms (*Figure 2A*). Zebrafish *trib2* is smaller than human *TRIB2* and mouse *Trib2* at 18.84 kb and only possesses two coding exons (*Figure 2A*). Human *TRIB3*, mouse *Trib3*, and zebrafish *trib3* share a similar exon organisation with a small non-coding first exon, followed by three coding exons (*Figure 2A*). Human *TRIB* 1, 2, and 3 are found on chromosomes 8, 2, and 20, respectively, while zebrafish *trib* 1, 2, and 3 are found on chromosomes 19, 20, and 21, respectively. Recently described human-specific variable number tandem repeats in the TRIB3 promotor region (*Örd et al., 2020*) were not identified in 50 kb of the zebrafish promotor region (sequence interrogated was TTg/aCATCA), suggesting potential differences in transcriptional/enhancer regulation, although these even differ between primates and have yet to be fully defined in other mammalian systems (*Örd et al., 2020*).

Homology between Tribbles isoforms across species is not only observed at the genetic level, but also at the protein level (*Hegedus et al., 2006*). Tribbles have three key protein domains: an N terminal PEST domain, a pseudokinase domain, and a functional C terminal (*Hegedus et al., 2007*). The pseudokinase contains a substrate-binding site within its catalytic loop, and the functional C terminus contains two binding sites for either MEK or COP enzymes (*Qi et al., 2006*; *Yokoyama et al., 2010*). These three binding sites were compared across human, mice, and zebrafish using the NCBI BLAST Global align online tool (*Figure 2B*). The pseudokinase catalytic loops in all three Tribbles family proteins (TRIB1–3), are found in human, mouse, and zebrafish. In the case of TRIB1 and TRIB2, there is no variation in the amino acid sequence of the pseudokinase catalytic loop across the three species (*Figure 2B*). The pseudokinase catalytic loop of both mouse TRIB3 and zebrafish Trib3 differs slightly from human TRIB3 with two amino acids that are different in mouse TRIB3 and one amino acid difference is observed in zebrafish Trib3. The amino acid sequences of human and zebrafish Tribbles were compared using the NCBI global align tool. Zebrafish Trib1 had the highest percentage identity when compared with human TRIB1 (52%), but also shared sequence homology with human TRIB2 with the highest identities match (66%) (*Figure 2C*). Zebrafish Trib2 shared the highest percentage identity with human TRIB2 (47% and 54%, respectively). Zebrafish Trib3 had high identity matches for both human TRIB2 and TRIB3 (68% and 64%, respectively) (*Figure 2C*). The overall size of Tribbles proteins remains consistent between human and mouse isoforms, with both human and mice TRIB1 sized at 372 amino acids (aa), human and mouse TRIB2 sized at 343aa. Mouse TRIB3 is 4aa shorter than human TRIB3 (354aa compared to 358aa). The zebrafish Tribbles isoforms are generally smaller proteins compared to the human and mouse Tribbles, with zebrafish Trib1 23aa smaller (at 349aa), Trib2 136aa smaller (at 207aa), and Trib3 10aa (at 348aa) compared to the human TRIB isoforms (*Figure 2D*). Predicted Alphafold models of zebrafish sequences of Trib1 and Trib3 overlayed with human protein predications demonstrate gross fold and alpha helix similarities (*Figure 2E, G*).

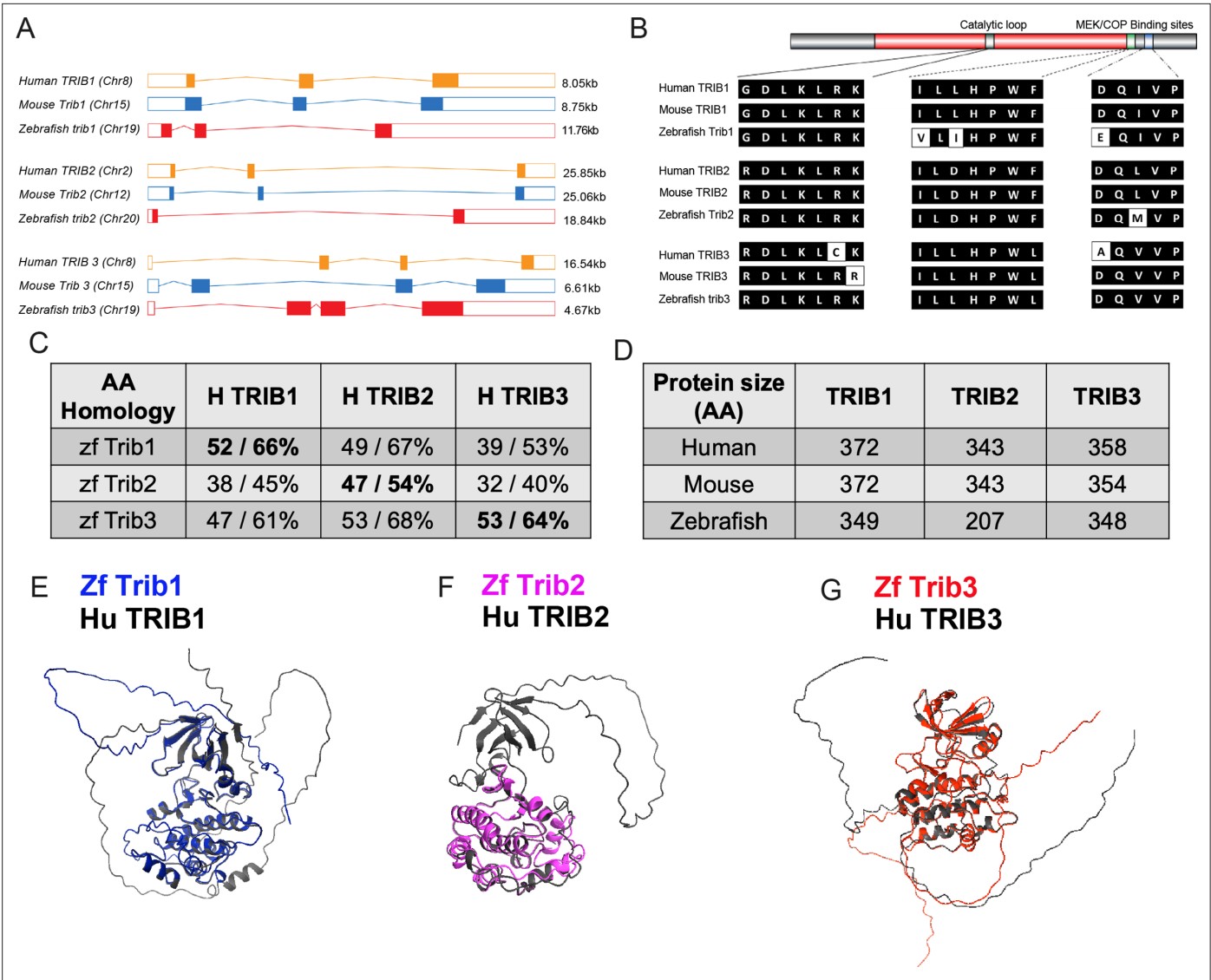

**Figure 2.** Zebrafish Tribbles share homology with their human and mice counterparts. (**A**) The gene organisation of human (orange) TRIB1, mouse (blue) Trib1, and zebrafish (red). Exon maps produced from Ensembl database. Chromosome number location (chr) and transcript sizes in kilobases (kb) are shown. (**B**) Comparison of the three catalytic domains of Tribbles; the pseudokinase catalytic loop and mitogen-activated protein kinase kinase (MEK)/ constitutive photomorphogenic 1 (COP1) bind sites, reveals high homology between species. (**C**) NCBI BLAST Global align revealed a high amino acid (AA) homology between zebrafish (zf) and human Tribbles protein sequences. Values described are positives/identities. (**D**) Protein sizes of the first and largest protein-coding transcript of each gene are depicted in the number of AA, values obtained from Ensembl and Uniprot databases. (**E–G**) Protein overlays of predicted structures (AlphaFold DB) of zebrafish (coloured) and human (black) sequences.

The online version of this article includes the following source data for figure 2:

**Source data 1.** ChimeraX (CXS) file for *Figure 2E*.

**Source data 2.** ChimeraX (CXS) file for *Figure 2F*.

**Source data 3.** ChimeraX (CXS) file for *Figure 2G*.

The Trib2 protein overlay was less well conserved, reflective of the smaller amino acid sequence of zebrafish Trib2 compared to human (*Figure 2F*).

To characterise the localisation of *trib* expression across the zebrafish larvae, whole-mount in situ hybridisation probes were developed for each zebrafish *trib* isoform. All *tribbles* isoforms showed highest expression in the brain of the developing zebrafish larvae at 3 dpf compared to sense probe controls (*Figure 3*). Expression of *trib* isoforms in immune cells was not detected by whole-mount in situ hybridisation of unchallenged larvae, compared to the expression of the highly expressed

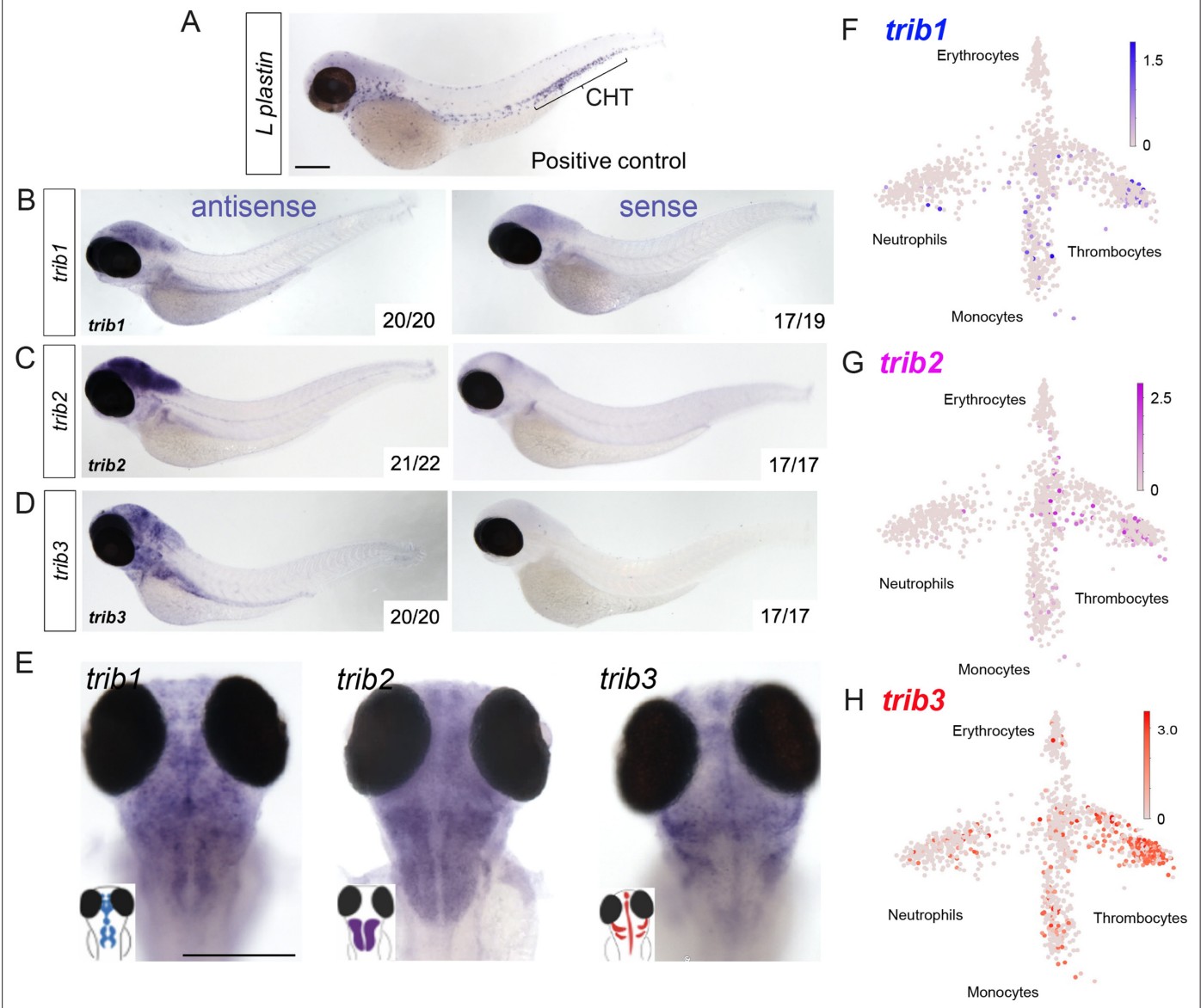

**Figure 3.** Expression of zebrafish *tribbles* are located in the brain in larvae and in immune cell subpopulations in adults. (**A**) Whole-mount in situ hybridisation of the pan-leukocyte marker *L-plastin* was used as a positive control to stain leukocytes, present predominantly in the caudal hematopoietic tissue (CHT). Scale bar = 200 µm. (**B**) Representative images of antisense and sense (control) whole-mount in situ hybridisation of *tribbles1*. Numbers correspond to number of fish with depictured pattern out of total fish imaged. *N* = 19–20 fish. (**C**) Antisense and sense (control) whole-mount in situ hybridisation of *tribbles2*. *N* = 17–22 fish. (**D**) Antisense and sense (control) whole-mount in situ hybridisation of *tribbles3*. *N* = 17–20 fish. (**E**) Magnified dorsal views (×8 magnification) showing distinct expression of each *trib* isoform in the brain (representative images from each group shown). Scale bar = 200 µm. (**F–H**) Gene expression of adult zebrafish leukocytes determined using the Zebrafish Blood Atlas (*Athanasiadis et al., 2017*). Each point represents a separate scRNAseq sample (cell); replicates performed across multiple zebrafish wildtype and transgenic strains. Each arm of schematic indicates a separate blood cell population (labelled). Deeper colour indicates higher expression (log10 scale bars described for each gene).

immune gene *l-plastin* (***Figure 3A–E***). However, this does not negate low, sub-threshold, levels of *tribbles* isoforms in immune cells. *trib* levels in blood cell lineages were assessed using the Zebrafish Blood Atlas (***Athanasiadis et al., 2017***; https://scrnaseq.shinyapps.io/scRNAseq_blood_atlas/), based on scRNAseq of adult zebrafish leukocytes. All *trib* isoforms were expressed in subpopulations of neutrophils, monocytes, and thrombocytes. *trib3* was expressed more abundantly and in a larger number of single-cell RNAseq samples than other *trib* isoforms, and was found in macrophages, neutrophils, and thrombocytes (***Figure 3F–H***).

In summary, zebrafish, mouse, and human Tribbles share sequence similarity and have similar gene organisation and conserved catalytic binding sites, making zebrafish a viable model to explore a physiological role for human Tribbles in mycobacterial infections. Zebrafish express *trib* isoforms in immune cell subpopulations in resting conditions, suggestive of roles in regulating innate immunity.

## Overexpression of *trib1* is host protective in a zebrafish mycobacteria infection model

To better understand how Tribbles can influence innate immunity and infection, genetic tools were generated to manipulate expression of zebrafish *trib* isoforms. Overexpression of zebrafish *tribbles* isoforms was achieved by injection of RNA at the one-cell stage. Injection of either *trib1*, *trib2*, or *trib3* RNA did not grossly affect larval development, with embryos developing with no obvious adverse effects (*Figure 4—figure supplement 1A, B*). To determine outcomes in infection, a zebrafish *Mycobacterium marinum* (*Mm*) larval model was used, in which *trib* RNAs were injected at the one-cell stage, leading to ubiquitous overexpression. Overexpression of *trib1* significantly decreased bacterial burden of *Mm* by approximately 50% (p < 0.001 compared to the vehicle control, phenol red PR; *Figure 4A, B*). Dominant active *hif-1α* (DA1, an RNA shown to significantly reduce *Mm* burden by ~50% *Elks et al., 2013*) was used as a positive RNA control with dominant negative *hif-1α* (DN1, an RNA shown to have no significant effect on *M. marinum* burden) used as a negative RNA control (*Elks et al., 2013*). Overexpression of *trib2* also significantly reduced bacterial burden compared to the PR control, but not to the same extent as the positive DA *hif-1α* control nor *trib1* overexpression (*Figure 4A, B*). In contrast, overexpression of *trib3* had no significant effect on the levels of bacterial burden compared to the vehicle PR control (*Figure 4A, B*). Together, these data demonstrate that overexpression of *trib1* has the strongest host-protective effect compared to overexpression of other *trib* isoforms, reducing *M. marinum* burden by approximately 50%.

*trib* knockdown/knockout tools were developed using CRISPR–Cas9 technology. Guide RNAs for each *trib* isoform were designed targeting the first coding exon of each *trib* gene and were injected into one-cell stage embryos, with *tyrosinase* (a control CRISPant which has negligible effects on innate immunity; *Isles et al., 2019*) CRISPant as a negative control. CRISPant efficiency was tested using PCR and restriction enzyme digest, with successful CRISPants disrupting the restriction site. Efficient guide RNAs were developed for both *trib1* and *trib3* (*Figure 4—figure supplement 2A, B*), however guide RNAs for *trib2* did not cause efficient knockdown. *Trib1* CRISPants had a higher burden of *Mm* compared to *tyrosinase* and *trib3* CRISPants (*Figure 4C, D*). To confirm the *trib1* knockdown phenotype, a stable CRISPR–Cas9 mutant for *trib1* was developed. Genotyping of F1 *trib1* CRISPR–Cas9 mutants revealed a 14-bp deletion (AGCAGATGTCCGCG) at the start of exon 1 (*Figure 4—figure supplement 3A–C*). This 14 bp deletion is predicted to create a truncated protein that is 94aa in length compared to the full-length 349aa wildtype protein, lacking the catalytic loop in the pseudokinase domain (*Figure 4—figure supplement 3D*) assuming no nonsense-mediated decay. Heterozygous *trib1*$^{+/-}$ were in-crossed and larvae were infected with *Mm* in the caudal vein at 1 dpi, imaged at 4 dpi and subsequently genotyped. At 4 dpi *Mm* infected *trib1*$^{-/-}$ mutants had significantly higher levels of bacterial burden compared to *trib1*$^{+/+}$ wildtype siblings (*Figure 4E*, *Figure 4—figure supplement 3E*).

## *trib1* manipulation does not affect total leukocyte numbers in larvae

TRIB1 has previously been shown to affect immune cell differentiation, with full-body *Trib1*-deficient mice possessing a greater number of neutrophils and a reduced number of anti-inflammatory macrophages compared to wildtype (*Satoh et al., 2013*). Zebrafish *trib* isoforms 1 and 3 were manipulated in neutrophil and macrophage transgenic reporter lines *Tg(mpx:GFP)i114* and *Tg(mpeg:nlsclover)sh436* and whole-body fluorescent cell counts were performed at 2 dpf to assess whether *trib* manipulation influenced zebrafish leukocyte number. Neither *trib1/trib3* overexpression nor *trib1/trib3* CRISPant knockdown grossly affected neutrophil or macrophage numbers (*Figure 5A–D*). Due to neutrophil numbers being most affected in the *Trib1*-deficient mice, we further investigated at the later timepoint of 3 dpf to determine whether neutrophil differentiation was affected and found that *Tg(mpx:GFP)i114*-positive neutrophil numbers were unchanged with *trib1* overexpression (*Figure 5E*) and *trib1* CRISPant knockdown (*Figure 5F*). We confirmed these data using a second neutrophil reporter transgenic line *Tg(lyz: nfsB.mCherry)sh260* with *trib1* overexpression (*Figure 5G*) and *trib1*

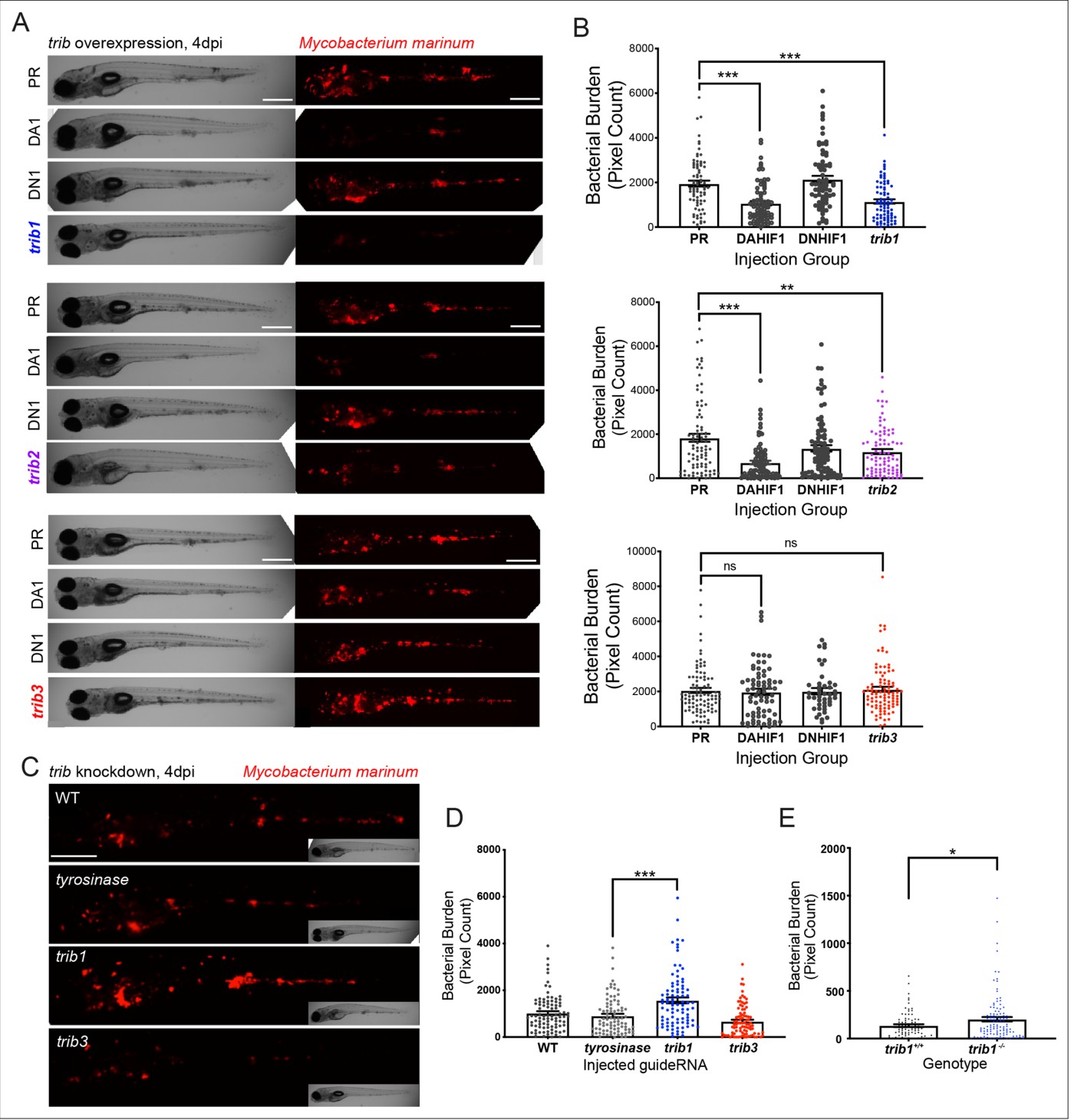

**Figure 4.** *trib1* overexpression is host protective against *Mm* infection. (**A**) Stereo-fluorescence micrographs of *Mm* mCherry infected 4 dpi larvae after injection at the single-cell stage with dominant active *hif-1α* (DA1), dominant negative *hif-1α* (DN1), *trib1, trib2, trib3* RNAs and phenol red (PR) as a negative vehicle control. DA1 and DN1 are RNA controls with DA1 having previously been shown to reduce infection levels (***Elks et al., 2013***). Scale bar = 200 µm. (**B**) Bacterial burden of larvae shown in (**A**). Data shown are mean ± standard error of the mean (SEM), *n* = 76–77 in *trib1*, *n* = 86–89 in *trib2*, and *n* = 43–95 in *trib3*, accumulated from three independent experiments for each *trib* gene. Statistical significance determined via one-way analysis of variance (ANOVA) with Bonferroni's multiple comparisons. p values shown are: \*\*p < 0.01 and \*\*\*p < 0.001. (**C**) Stereo-fluorescence micrographs of Mm mCherry infected 4 dpi larvae after injection with *tyrosinase* (control), *trib1* and *trib3* CRISPR guides (CRISPants). Scale bar = 200 µm. (**D**) Bacterial burden of larvae shown in (**C**). Data shown are mean ± SEM, *n* = 87–90 fish accumulated from three independent experiments. Statistical significance determined via one-way ANOVA with Bonferroni's multiple comparisons. p values shown are: \*\*\*p < 0.001. (**E**) Bacterial burden of *trib1*−/− stable

*Figure 4 continued on next page*

*Figure 4 continued*

knockout larvae compared to wildtype (*trib1*[+/+]) siblings. Data shown are mean ± SEM, *n* = 82–114 fish accumulated from four independent experiments. Statistical significance determined via an unpaired *t*-test. p values shown are: *p < 0.05.

The online version of this article includes the following source data and figure supplement(s) for figure 4:

**Source data 1.** Numerical data for the graph of *Figure 4B trib1*.

**Source data 2.** Numerical data for the graph of *Figure 4B trib2*.

**Source data 3.** Numerical data for the graph of *Figure 4B trib2*.

**Source data 4.** Numerical data for the graph of *Figure 4D*.

**Source data 5.** Numerical data for the graph of *Figure 4E*.

**Figure supplement 1.** *tribbles* RNA overexpression caused no observable developmental defects.

**Figure supplement 2.** CRISPR guide RNAs against *trib1* and *trib3* efficiently disrupt a restriction digest site.

**Figure supplement 2—source data 1.** Raw unedited gel image of *Figure 4—figure supplement 2A*.

**Figure supplement 2—source data 2.** PDF containing *Figure 4—figure supplement 2A* and the original gel image with highlighted bands and samples labels.

**Figure supplement 2—source data 3.** Raw unedited gel image of *Figure 4—figure supplement 2B*.

**Figure supplement 2—source data 4.** PDF containing *Figure 4—figure supplement 2B* and the original gel image with highlighted bands and samples labels.

**Figure supplement 3.** Generation of a *tribbles1* stable mutant line.

**Figure supplement 3—source data 1.** Raw unedited gel image of *Figure 4—figure supplement 3B*.

**Figure supplement 3—source data 2.** PDF containing *Figure 4—figure supplement 3B* and the original gel image with highlighted bands and samples labels.

**Figure supplement 3—source data 3.** Numerical data for the graph of *Figure 4—figure supplement 3G*.

CRISPant knockdown (*Figure 5H*) showing no differences in numbers at 2 and 3 dpf. These data suggest that the host-protective effect of *trib1* overexpression is not due to an increase in number of innate immune cells.

## *trib1* overexpression increases production of proinflammatory factors

Zebrafish *trib1* manipulation had a profound effect on host–pathogen interaction, with overexpression reducing Mm bacterial burden and CRISPant knockdown/CRISPR mutant increasing burden. We therefore investigated the roles of *trib1* manipulation on the innate immune system. To investigate whether *trib1* influenced the inflammatory profiles of zebrafish leukocytes, production of the proinflammatory factors, *il-1β* and NO were measured using a combination of transgenic reporter lines and immunostaining. Overexpression of *trib1* increased the levels of *il1β:GFP* (in a *Tg(il-1β:GFP)sh445* reporter line), to similar levels as the positive control DA Hif-1α, compared to PR injected controls (*Figure 6A, B*). *trib3* overexpression did not increase levels of *il1β:GFP* and levels were similar to the negative controls DN Hif-1α and PR (*Figure 6A, B*). Similarly, *trib1* overexpression increased the levels of anti-nitrotyrosine staining (a proxy for immune cell antimicrobial NO production; *Forlenza et al., 2008*) compared to PR controls, to similar levels as DA Hif-1α (*Elks et al., 2014*; *Elks et al., 2013*; *Figure 6C, D*). *trib3* overexpression did not increase levels of proinflammatory nitrotyrosine (*Figure 6C, D*).

## *Trib1* overexpression does not activate Hif signalling

Tribbles regulation has been mechanistically linked to hypoxia and HIF-1α in a variety of cancer cells (*Hong et al., 2019*; *Wennemers et al., 2011*; *Xing et al., 2020*) and in *Drosophila* fat body tissue (*Noguchi et al., 2022*). Due to the host-protective effect of *trib1* overexpression in Mm infection closely mimicking that of DA-Hif1α, a potential mechanistic link between the *hif-1α* and *trib1* pathways was investigated. *trib1* and *trib3* were overexpressed in a Hif-α transgenic reporter line, *Tg(phd3:GFP) i144* (*phd3* is a downstream target of Hif-α signalling) (*Santhakumar et al., 2012*). Neither *trib1* nor *trib3* overexpression activated the *phd3:GFP* line to detectable levels, indicating that *trib1* overexpression is not substantially increasing Hif-1α signalling to mediate *Mm* control (*Figure 7A, B*). These data suggest that the protective effects of *trib1* act via a different mechanism than Hif-1α activation.

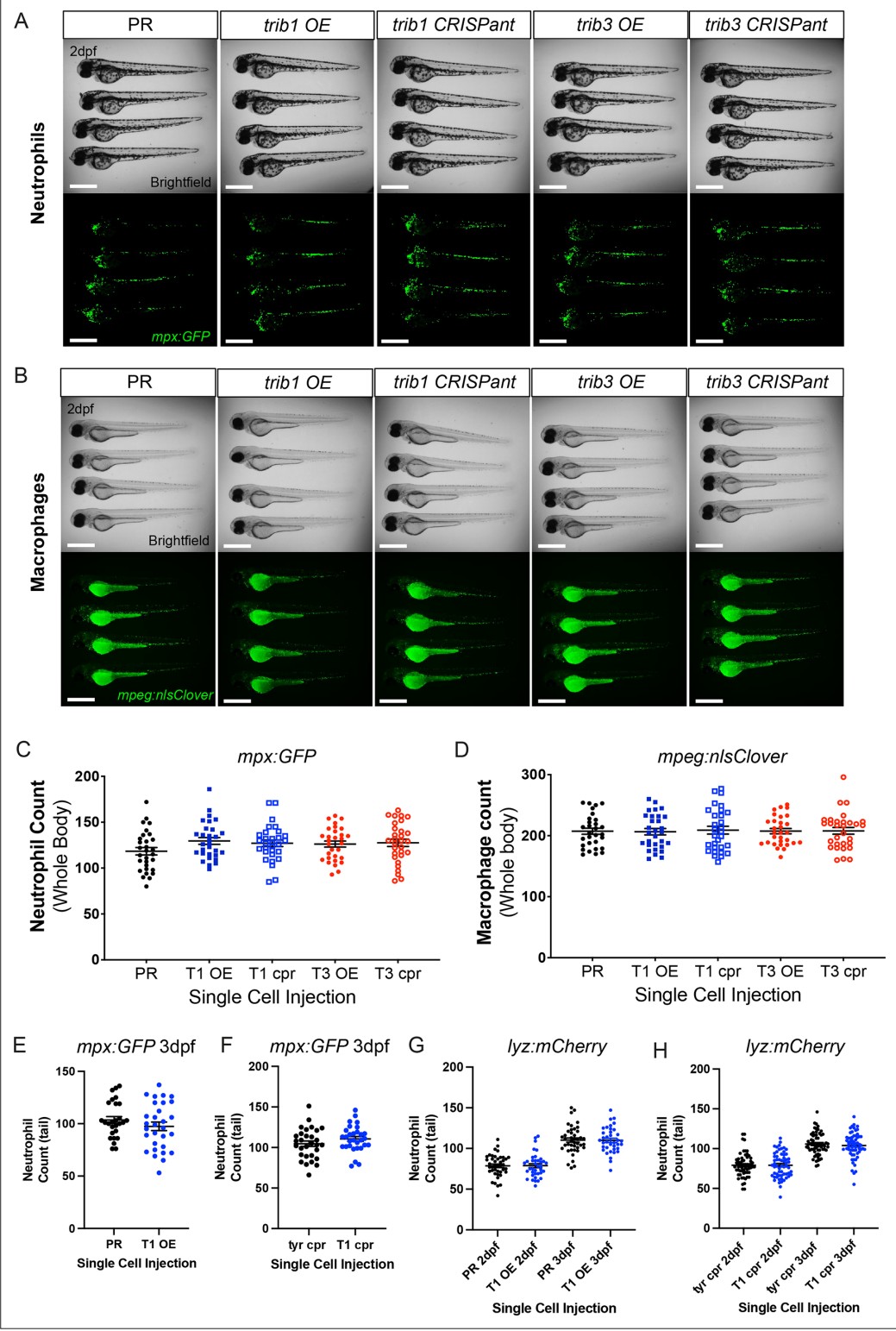

**Figure 5.** *trib* manipulation does not affect leukocyte number in zebrafish larvae. (**A**) Example brightfield and fluorescent micrographs of neutrophils (*Tg(mpx:GFP)i114*) in four 2 dpf larvae injected at the one-cell stage with either phenol red (PR), *trib1* RNA overexpression (OE), *trib1* CRISPant, *trib3* RNA overexpression (OE), or *trib3* CRISPant. Scale bars = 200 μM. (**B**) Example brightfield and fluorescent micrographs of macrophages (*Tg(mpeg:nlsclover)sh436*) in four 2 dpf larvae injected at the one-cell stage with either PR, *trib1* RNA overexpression (OE), *trib1* CRISPant, *trib3* RNA overexpression (OE), or *trib3* CRISPant. Scale bars = 200 μM.

*Figure 5 continued on next page*

*Figure 5 continued*

(**C**) Wholebody neutrophil numbers (*Tg(mpx:GFP)i114*) in 2 dpf larvae injected at the one-cell stage with either PR, *trib1* RNA overexpression (T1 OE), *trib1* CRISPant (T1 cpr), *trib3* RNA overexpression (T3 OE), or *trib3* CRISPant (T3 cpr). Data shown are mean ± standard error of the mean (SEM), *n* = 30 fish per group. (**D**) Wholebody macrophage numbers (*Tg(mpeg:nlsclover)sh436*) in 2 dpf larvae injected at the one-cell stage with either PR, *trib1* RNA overexpression (T1 OE), *trib1* CRISPant (T1 cpr), *trib3* RNA overexpression (T3 OE), or *trib3* CRISPant (T3 cpr). Data shown are mean ± SEM, *n* = 30 fish per group. (**E**) Total tail neutrophil numbers (*Tg(mpx:GFP)i114*) in 3 dpf larvae injected at the one-cell stage with either PR or *trib1* RNA overexpression (T1 OE). Data shown are mean ± SEM, *n* = 30 fish per group. (**F**) Total tail neutrophil numbers (*Tg(mpx:GFP)i114*) in 3 dpf larvae injected at the one-cell stage with either *tyrosinase* control CRISPant (tyr cpr) or *trib1* CRISPant (T1 cpr). Data shown are mean ± SEM, *n* = 30 fish per group. (**G**) Total tail neutrophil numbers (*Tg(lyz: nfsB.mCherry)sh260*) in 2 and 3 dpf larvae injected at the one-cell stage with either PR or *trib1* RNA overexpression (T1 OE). Data shown are mean ± SEM, *n* = 37–42 fish per group. (**H**) Total tail neutrophil numbers (*Tg(lyz: nfsB.mCherry)sh260*) in 2 and 3 dpf larvae injected at the one-cell stage with either *tyrosinase* control CRISPant (tyr cpr) or *trib1* CRISPant (T1 cpr). Data shown are mean ± SEM, *n* = 60 fish per group.

The online version of this article includes the following source data for figure 5:

**Source data 1.** Numerical data for the graph of *Figure 5C*.

**Source data 2.** Numerical data for the graph of *Figure 5D*.

**Source data 3.** Numerical data for the graph of *Figure 5E*.

**Source data 4.** Numerical data for the graph of *Figure 5F*.

**Source data 5.** Numerical data for the graph of *Figure 5G*.

**Source data 6.** Numerical data for the graph of *Figure 5H*.

## The host-protective effect of *trib1* is dependent on *cop1*

An important binding partner of the TRIB1 protein is the E3 ubiquitin ligase, COP1 (*Jamieson et al., 2018*; *Kung and Jura, 2019*; *Murphy et al., 2015*). To investigate whether the host-protective effects of *trib1* overexpression in *Mm* infection were *cop1*-mediated, a *cop1* CRISPant was generated. The zebrafish *cop1* gene (ENSDARG00000079329) is located on the forward strand of chromosome 2 and has 20 exons, all of which are coding (*Figure 8—figure supplement 1A, B*). It has a single coding transcript, producing a Cop1 protein of 694 amino acids. The zebrafish *cop1* gene shares synteny and conserved sequence with both the human COP1 and murine Cop1 (determined using the ZFIN database https://zfin.org/). Cop1 mutant larvae appeared normal with no gross developmental defect and had no change in wholebody neutrophil number at 2 dpf (*Figure 8—figure supplement 1C*).

In order to investigate whether the protective effect of *trib1* overexpression is *cop1*-mediated, *trib1* overexpression was combined with *cop1* CRISPants in *Mm* infected larvae. Overexpression of *trib1* significantly reduced bacterial burden compared to PR controls when co-injected with *tyrosinase* guide (*Figure 8A, B*). The bacterial burden of *cop1* CRISPants was not significantly different to the tyrosinase control group nor the tyrosinase control with *trib1* overexpression group. When *trib1* was overexpressed in *cop1* CRISPants, there was no significant decrease in burden, with the protective effect of *trib1* lost (*Figure 8A, B*) indicating that the protective effect of *trib1* overexpression is dependent on *cop1*.

The effect of *cop1* knockdown on the production of antimicrobial NO production was investigated using the anti-nitrotyrosine antibody. Overexpression of *trib1* significantly increased neutrophil anti-nitrotyrosine fluorescence levels compared to the PR control in the *tyrosinase* controls (*Figure 8C, D*). The *cop1* CRISPant group possessed comparable anti-nitrotyrosine levels to both the PR and tyrosinase control groups. *trib1* overexpression in the *cop1* CRISPants did not increase anti-nitrotyrosine levels and instead was comparable with the *cop1* CRISPants alone and both PR and tyrosinase controls (*Figure 8C, D*).

Together these data show that when *cop1* is knocked down the antimicrobial and host-protective effects of *trib1* overexpression are lost, indicating a dependency of the *trib1* effect on *cop1*.

## Discussion

TRIB1 has previously been shown to be a key regulator of multiple inflammatory factors and immune cell function, influencing pathologies with an inflammatory component including cancer and

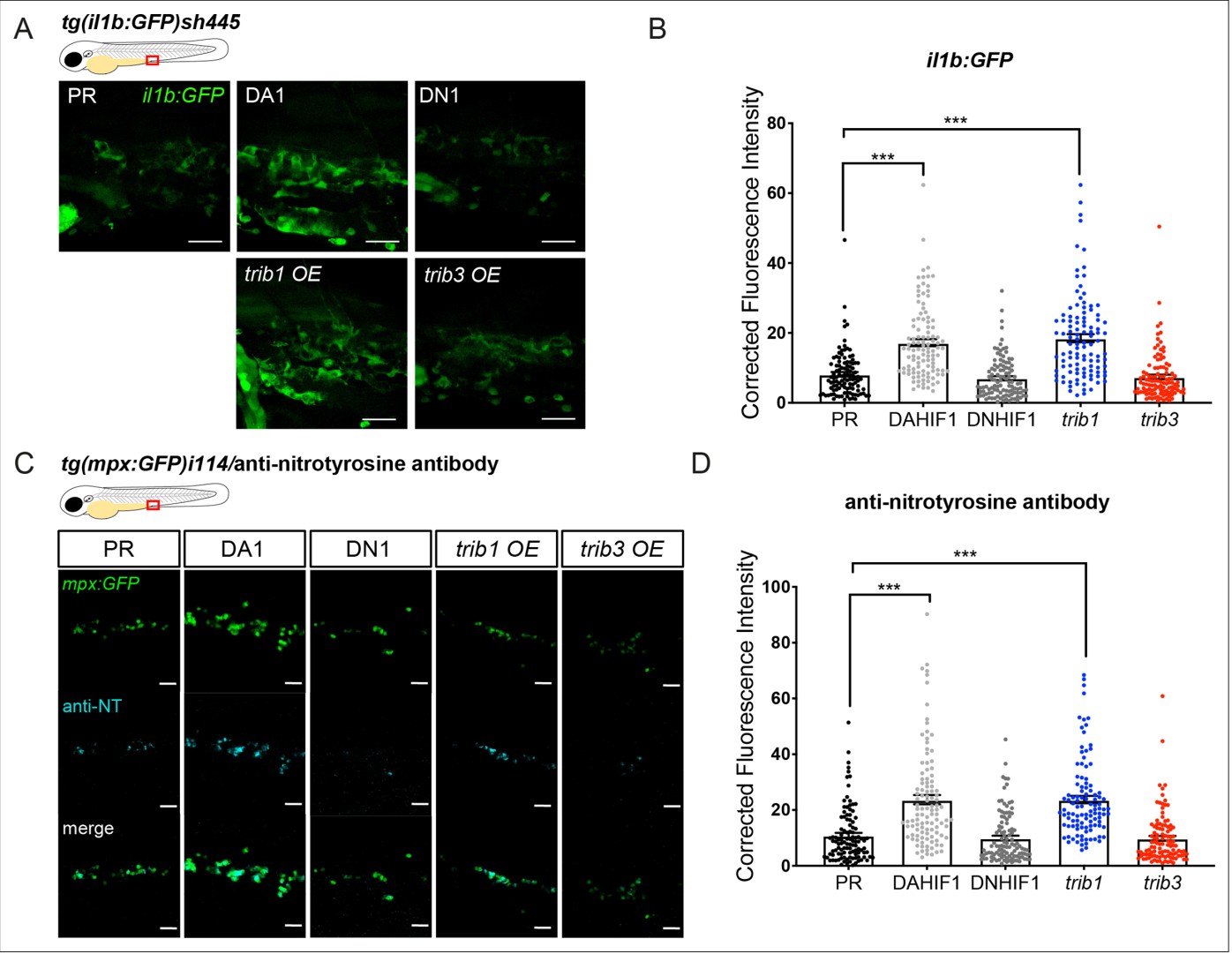

**Figure 6.** *trib1* overexpression increases production of proinflammatory *il-1β* and nitrotyrosine in the absence of infection. (**A**) Fluorescent confocal micrographs of 2 dpf caudal vein region of *TgBAC(il-1β:eGFP)sh445* transgenic larvae. *il-1β:GFP* expression was detected by GFP levels. Larvae were injected at the one-cell stage with dominant negative (DN) or dominant active (DA) Hif-1α or phenol red (PR) controls and *trib1* and *trib3* test RNAs. Scale bars = 25 µm. (**B**) Corrected fluorescence intensity levels of *il-1β:GFP* confocal z-stacks in uninfected larvae at 2 dpf of data shown in (**A**). Dominant active Hif-1α (DA1) controls and *trib1* fish had significantly increased *il-1β:GFP* levels in the absence of Mm bacterial challenge compared to PR and dominant negative Hif-1α (DN1) injected controls and *trib3* RNA injected embryos. Data shown are mean ± standard error of the mean (SEM), *n* = 108 cells from 18 embryos accumulated from 3 independent experiments. Statistical significance was determined using one-way analysis of variance (ANOVA) with Bonferroni's multiple comparisons post hoc test. p values shown are: ***p < 0.001. (**C**) Fluorescence confocal z-stacks of the caudal vein region of 2 dpf *mpx:GFP* larvae (neutrophils) immune labelled with anti-nitrotyrosine (cyan) in the absence of Mm infection. Larvae were injected at the one-cell stage with dominant negative (DN) or dominant active (DA) Hif-1α or PR controls and *trib1* and *trib3* test RNAs. Scale bars = 25 µm. (**D**) Corrected fluorescence intensity levels of anti-nitrotyrosine antibody confocal z-stacks shown in (**C**). Data shown are mean ± SEM, *n* = 108 cells from 18 embryos accumulated from 3 independent experiments. Statistical significance was determined using one-way ANOVA with Bonferroni's multiple comparisons post hoc test. p values shown are: ***p < 0.001.

The online version of this article includes the following source data for figure 6:

**Source data 1.** Numerical data for the graph of *Figure 6B*.

**Source data 2.** Numerical data for the graph of *Figure 6D*.

atherosclerosis (*Johnston et al., 2015*). Innate immunity and production of inflammatory factors are important defence mechanisms against invading pathogens, yet the role of Tribbles in the immune response to infection is poorly understood. Here, we provide evidence that mycobacterial antigen stimulation both in vitro and in vivo induces human TRIB1 and TRIB2 expression independent of TB

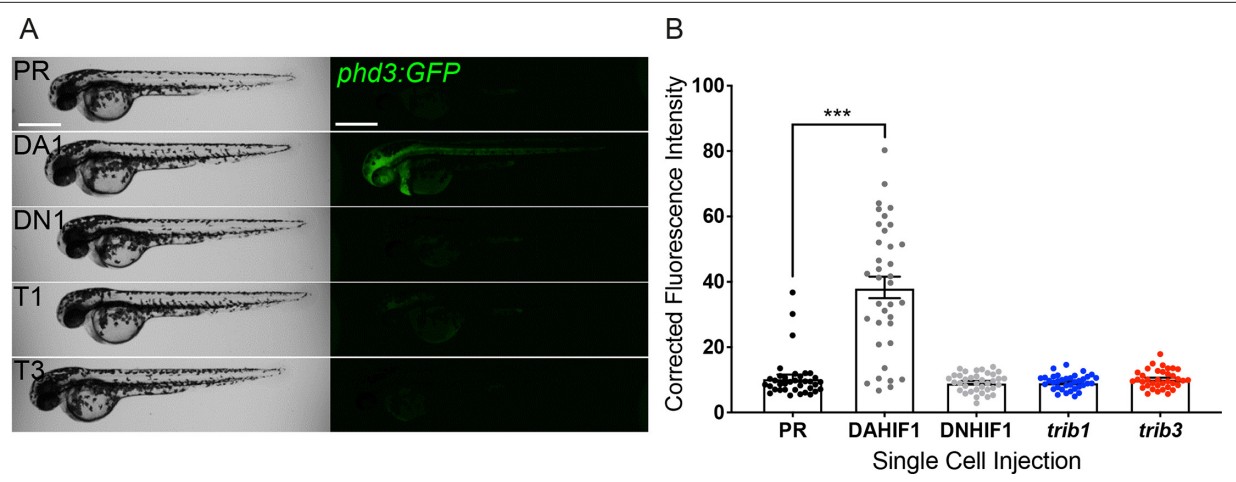

**Figure 7.** *trib1* and *trib3* overexpression do not induce expression of the Hif reporter *phd3:GFP*. (**A**) Stereo-fluorescence micrographs of 2 dpf *phd3:GFP* larvae injected with phenol red (PR), dominant active *hif-1α* (DA1) and dominant negative Hif-1α (DN1) RNA controls alongside *trib1* (T1) and *trib3* (T3) RNA. Scale bars = 200 μM. (**B**) Corrected fluorescence intensity levels of *phd3:GFP* (Hif reporter) larvae shown in (**A**). Data shown are mean ± standard error of the mean (SEM), *n* = 30 embryos accumulated from three independent experiments. Statistical significance determined through one-way analysis of variance (ANOVA) with multiple comparisons. p values shown are: ***p < 0.001.

disease status. We used a zebrafish TB model to show that *trib1* expression has functional roles in the innate immune response to infection in vivo. trib1 is required to control Mm infection, associated with increased production of the antimicrobial factors *il-1β* and NO. We also show a role for *cop1*, a binding partner of TRIB1, which is required for the host-protective effects of *trib1* overexpression. The in vivo tools, developed here to investigate the immune roles of *tribbles* in zebrafish, create new opportunities to further investigate Tribbles1 as a potential therapeutic target, not only in infection, but in a wider range of disease contexts that have an innate immunity component.

For the first time, we have identified that Tribbles1 is important in the host response to mycobacterial infection, with *TRIB1* being upregulated in human monocytes after mycobacterial antigen challenge and overexpression of *trib1* being host-protective in a zebrafish TB model. TRIB1 is a well-known regulator of innate immune cells and functions in inflammatory settings playing a key role in the regulation of proinflammatory profiles (*Arndt et al., 2018*; *Niespolo et al., 2020*; *Ostertag et al., 2010*). Trib1⁻/⁻ mice have a defective inflammatory response, with reduced proinflammatory gene expression (including *Nos2* and *Il-1β* compared to controls) resulting in bone-marrow-derived macrophages (BMDMs) that produce lower NO and have defective phagocytosis (*Arndt et al., 2018*). In zebrafish, *trib1* overexpression increased production of proinflammatory factors, indicating that similar control of NO and Il-1β by TRIB1 is conserved in fish, although NO was primarily observed in neutrophils rather than macrophages matching previous observations in the zebrafish model (*Elks et al., 2014*; *Elks et al., 2013*).

TRIB1 has been associated with inflammation and immune response, whereas TRIB3 is strongly associated with metabolic function, including the regulation of glucose homeostasis (*Angyal and Kiss-Toth, 2012*; *Prudente et al., 2012*; *Zhang et al., 2013*) which also has regulatory roles in innate immune cells such as macrophages (*Steverson et al., 2016*; *Wang et al., 2012*). There are robust links between glucose metabolism and innate immune responses, such as the glycolytic switch which is closely related to macrophage polarisation (*Zhu et al., 2015*). Immune cell glucose and lipid metabolism have important roles in infection defence. Lipid droplets form in macrophages during *Mtb* infection that are potentially used as source of lipids by *Mtb* to allow for intracellular growth (*Daniel et al., 2011*). However, more recent findings suggest that lipid droplets are formed during the immune activation process after macrophage *Mtb* infection (*Knight et al., 2018*), that can subsequently influence the dynamic response of macrophage host defence (*Menon et al., 2019*). In murine atheroma models, Trib1 increased the lipid accumulation in macrophages leading to the formation of foam cells (*Johnston et al., 2019*). In *Drosophila melanogaster*, *trbl* knockdown increased circulating triglyceride levels (*Das et al., 2014*) and in mice where TRIB3 knockdown in a murine adipose cell line (3T3-L1)

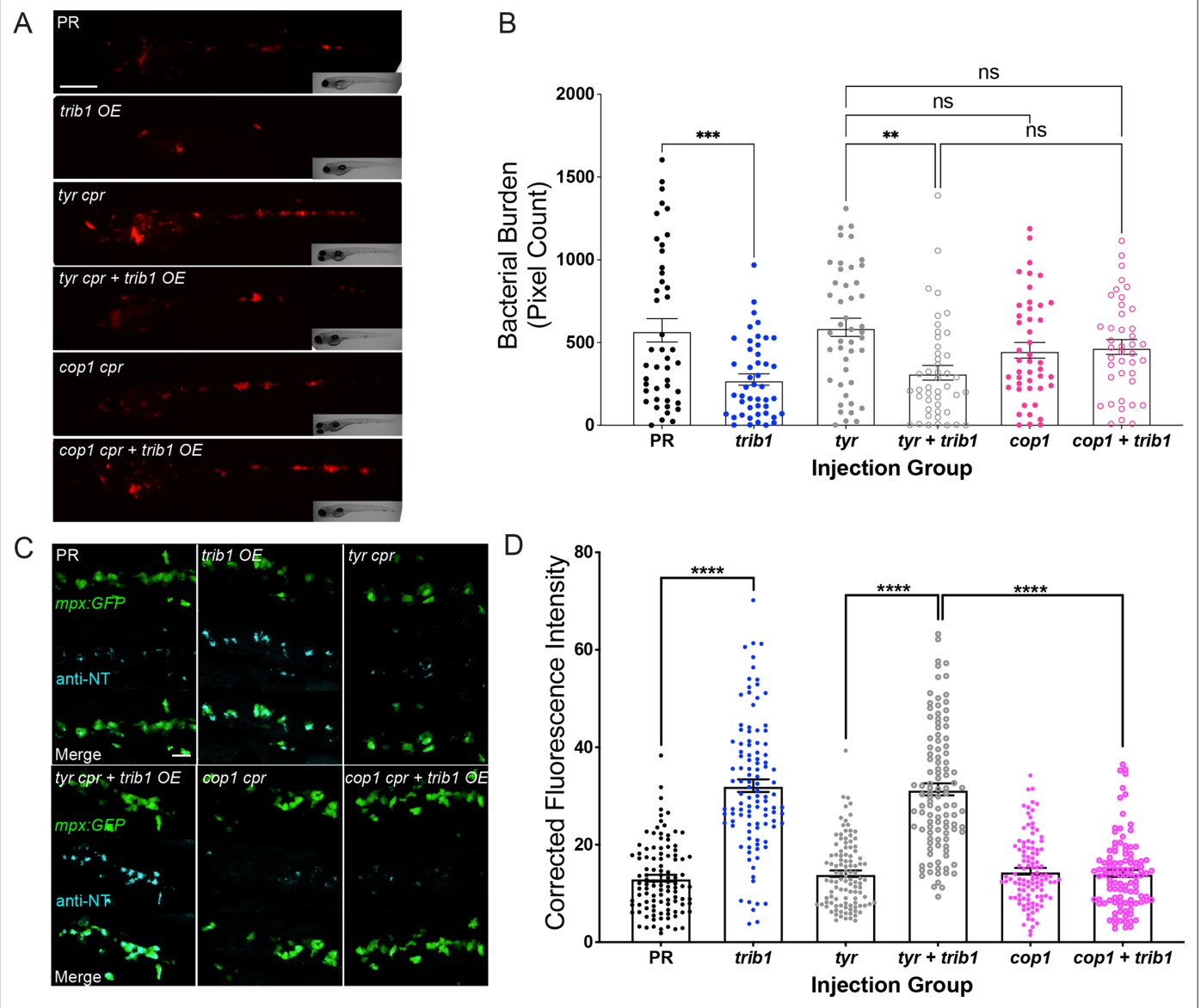

**Figure 8.** The host-protective effect of *trib1* overexpression requires *cop1*. (**A**) Stereo-fluorescence micrographs of Mm mCherry infected 4 dpi larvae after injection with *trib1* RNA (overexpression, OE) and *cop1* guide RNA (CRISPants, *cpr*) using phenol red (PR, vehicle) and *tyrosinase* (*tyr cpr*, unrelated guide RNA) CRISPants as negative controls. Scale bar = 200 μM. (**B**) Bacterial burden of larvae shown in (**A**). Data shown are mean ± standard error of the mean (SEM), *n* = 71–76 accumulated from three independent experiments. Statistical significance determined via one-way analysis of variance (ANOVA) with Bonferroni's multiple comparisons. p values shown are: **p < 0.01 and ***p < 0.001. (**C**) Fluorescence confocal z-stacks of the caudal vein region of 2 dpf *mpx:GFP* larvae (neutrophils) immune labelled with anti-nitrotyrosine (cyan) in the absence of Mm infection. Larvae were injected at the one-cell stage with *trib1* RNA (overexpression, OE) and *cop1* guide RNA (CRISPants, cpr) using PR (vehicle) and tyrosinase (unrelated guide RNA) CRISPants as negative controls. Scale bar = 30 μm. (**D**) Corrected fluorescence intensity levels of anti-nitrotyrosine antibody confocal z-stacks shown in (**C**). Data shown are mean ± SEM, *n* = 108 cells from 18 embryos accumulated from three independent experiments. Statistical significance was determined using one-way ANOVA with Bonferroni's multiple comparisons post hoc test. p values shown are: ****p < 0.0001.

The online version of this article includes the following source data and figure supplement(s) for figure 8:

**Source data 1.** Numerical data for the graph of *Figure 8B*.

**Source data 2.** Numerical data for the graph of *Figure 8D*.

**Figure supplement 1.** An efficient CRISPR–Cas9 guide RNA was generated for the zebrafish *cop1* gene.

**Figure supplement 1—source data 1.** Raw unedited gel image of *Figure 8—figure supplement 1B*.

**Figure supplement 1—source data 2.** PDF containing *Figure 8—figure supplement 1B* and the original gel image with highlighted bands and samples labels.

**Figure supplement 1—source data 3.** Numerical data for the graph of *Figure 8—figure supplement 1C*.

increased intracellular triglycerides (*Takahashi et al., 2008*) and targeted deletion of murine TRIB3 resulted in elevated triglyceride levels in the liver (*Örd et al., 2018*). Macrophage lipid metabolism and handling during *Mtb* infection could therefore potentially be influenced by Tribbles.

It is interesting to note that overexpression or knockdown of *trib* genes did not affect macrophage and neutrophil differentiation in zebrafish larvae. Trib1-deficient mice have been shown to have defects in the differentiation of a number of different immune cells, including an increase in neutrophils due to commitment of the granulocyte precursor lineage to neutrophils at the expense of eosinophils (*Mack et al., 2019*) and a decrease in M2-like tissue-resident macrophages (*Satoh et al., 2013*). The effects of Trib1 deficiency on eosinophil/neutrophil differentiation has been shown to be dependent on C/EBP transcription factors (*Mack et al., 2019*). Similar to mammalian neutrophil differentiation, zebrafish neutrophil differentiation is partially regulated by C/EBP transcription factors including Cebpα (*Dai et al., 2016*; *Yuan et al., 2011*), Cebp1 (the functional homolog of mammalian C/EBPε, *Kim et al., 2016*) and Cebpβ (*Wei et al., 2020*). It was therefore surprising that *trib1* manipulation did not affect neutrophil differentiation in the zebrafish and highlights potential differences between the function of zebrafish *trib1* compared to murine TRIB1. An important C/EBP in zebrafish myelopoiesis is a zebrafish-specific isoform called Cebp1 that is myeloid expressed (*Lyons et al., 2001*). This has a highly conserved carboxy-terminal bZIP domain with mammalian homologues, but the amino-terminal domains are unique. Reduction of Cebp1 did not ablate initial macrophage or granulocyte development in 24 hpf zebrafish larvae (*Lyons et al., 2001*). Recent findings demonstrate that Cebp1 controls eosinophilopoiesis in zebrafish larvae and has important roles in balancing the neutrophil and eosinophil lineages (*Li et al., 2024*), as in mice (*Mack et al., 2019*), however, eosinophils do not emerge until after 5 dpf beyond the stages in our study (*Balla et al., 2010*; *Li et al., 2024*). It remains unclear as to whether M2-like macrophages are affected by *trib1* modulation in zebrafish as they are in mice (*Satoh et al., 2013*), as biomarkers/transgenic tools to investigate the existence of M2-like macrophages in larvae have only recently been developed (*Hammond et al., 2023*), but could be of interest in future studies.

Many of the reported regulatory functions of TRIB1 in immune cells are dependent on COP1 (*Kung and Jura, 2019*). In our zebrafish Mm model the protective effect of *trib1* overexpression was reduced when *cop1* was depleted, demonstrating a requirement for *cop1* expression to improve the host response to infection. Interestingly, *cop1* CRISPants possess a subtly reduced burden compared to controls without the overexpression of *trib1*, suggesting that *cop1* depletion alone may offer a small level of protection. In cancer cell lines infected with *Mycobacterium bovis* Bacillus Calmette–Guérin (BCG), BCG-induced Sonic Hedgehog signalling increasing COP1 expression, leading to the inhibition of apoptosis in the cell line (*Holla et al., 2014*), indicating there may be a COP1 response to mycobacterial infection. The TRIB1/COP1 complex is responsible for the regulation of multiple targets, including C/EBPs (*Liu et al., 2013*; *Yoshida et al., 2013*). The lack of changes observed in neutrophil and macrophage numbers in *trib1*/*cop1* knockdowns in our zebrafish studies, discussed above, may be suggestive of C/EBP independent mechanisms. There is evidence suggesting that COP1 and C/EBP have distinct binding sites on TRIB1, potentially unlinking their activity in some biological situations and this may be the case in zebrafish haematopoeisis (*Murphy et al., 2015*). There are many other candidate pathways other than C/EBPs that Trib1 and Cop1 interactions can influence that could lead to the host-protective effect observed, with examples including mitogen-activated protein kinases (*Niespolo et al., 2020*), serine threonine kinases (*Durzynska et al., 2017*), JAK/STAT (*Arndt et al., 2018*), and beta-Catenin (*Zahid et al., 2022*), likely context and cell dependent. For example, the neutrophil NO response generated by *trib1* overexpression in zebrafish could be produced through JAK/STAT signalling which, in mice, regulates macrophage NO via TRIB1 control of STAT3 and STAT6 (*Arndt et al., 2018*). The zebrafish will be a useful in vivo model to investigate and dissect these pathways further.

If proinflammatory signals are initiated early in infection, this can reduce infection burden and improve infection outcomes. An example of this is control of the proinflammatory response by HIF-1α, where early stabilisation and activation of HIF-1α signalling is host beneficial in mycobacterial infection (*Elks et al., 2013*; *Lewis and Elks, 2019*; *Ogryzko et al., 2019*). The host-protective effect of *trib1* overexpression closely mimicked the effects of Hif-1α stabilisation, with an increase in the production of antimicrobial factors NO and *il-1β* and a decrease in *Mm* infection burden (*Elks et al., 2013*; *Ogryzko et al., 2019*). TRIB3 has been associated with HIF-1α in renal cell carcinoma patients with

HIF-1α binding to multiple regions in the TRIB3 promoter resulting in upregulation of TRIB3 expression (*Hong et al., 2019*). In lung adenocarcinoma cells, TRIB3 knockdown decreased levels of HIF-1α (*Xing et al., 2020*), indicating a feedback loop between TRIB3 and HIF-1α, where one can regulate the other. In common with TRIB3, a potential link between TRIB2 and HIF-1α has been reported, as depletion of TRIB2 significantly decreased the effect of TNF-α on HIF-1α stability and accumulation in multiple cancer cell lines (*Schoolmeesters et al., 2012*). However, there is no current link identified between TRIB1 and HIF-1α and this was reflected in our findings showing that *trib1* overexpression did not lead to an increase in a well-validated Hif-α reporter line (*Santhakumar et al., 2012*).

Together our findings show a potential therapeutic application of targeting Trib1 to improve infection outcomes. We demonstrate *il-1b* and NO control by Trib1, suggesting that Trib1 controls multiple immune pathways and that therapeutic Trib1 manipulation may be more effective than targeting individual immune pathways alone. Due to its potential functions in multiple pathways, any targeting of Trib1 must be carefully controlled. For example, overexpression of TRIB1 in chronic mycobacterial infections may be beneficial against infection, but could trigger immunopathology. The concept of host immunomodulation is an emerging therapeutic avenue for infectious disease, especially with the continually increasing problem of AMR in multiple pathogens, and could potentially be used alongside antimicrobial drug treatment. To aid the efficiency of host immunomodulation, and to help avoid off-target effects, specific targeting methods can be used. Polymersomes have been shown to be a promising avenue for drug delivery to immune cells and could be utilised for the delivery of host immunomodulatory compounds and factors (*Dal et al., 2020*). Further research into TRIB1 as a target for host-derived therapies could potentially improve infection outcome of mycobacterial infection via pharmacological targeted delivery methods and transient manipulation through genetic approaches.

# Materials and methods

## Key resources table

| Reagent type (species) or resource | Designation | Source or reference | Identifiers | Additional information |
|---|---|---|---|---|
| Gene (*Danio rerio*) | *TgBAC(mpx:EGFP)i114* | *Elks et al., 2011* | i114Tg RRID: ZFIN_ZDB-ALT-070118-2 | Transgenic |
| Gene (*Danio rerio*) | *Tg(lyz:nfsβ-mCherry)sh260* | *Buchan et al., 2019* | sh260Tg RRID: ZFIN_ ZDB-ALT-190925–14 | Transgenic |
| Gene (*Danio rerio*) | *Tg(mpeg:nlsclover)* | *Bernut et al., 2019* | sh436Tg RRID: ZFIN_ZDB-ALT-191219-4 | Transgenic |
| Gene (*Danio rerio*) | *Tg(mpeg1:mCherryCAAX)* | *Bojarczuk et al., 2016* | sh378Tg RRID: ZFIN_ZDB-ALT-160414–5 | Transgenic |
| Gene (*Danio rerio*) | *Tg(phd3:GFP)* | *Santhakumar et al., 2012* | sh144Tg RRID: ZFIN_ZDB-ALT-120925-1 | Transgenic |
| Gene (*Danio rerio*) | *TgBAC(il-1β:GFP)* | *Ogryzko et al., 2019* | Sh445Tg RRID: ZFIN_ZDB-ALT-190307-8 | Transgenic |
| Gene (*Danio rerio*) | *trib1^{sh628/sh628}* CRISPR mutant | This paper | sh628 | CRISPR mutant |
| Genetic reagent (*Danio rerio*) | *trib1* CRISPR–Cas9 guide RNA | This paper | *trib1* CRISPR | AGCCCGTGAGCAGATGTCCGCGG |
| Genetic reagent (*Danio rerio*) | *trib3* CRISPR–Cas9 guide RNA | This paper | *trib3* CRISPR | TCAACTCGCTTCAGTCGCAGTGG |
| Genetic reagent (*Danio rerio*) | *cop1* CRISPR–Cas9 guide RNA | This paper | *cop1* CRISPR | CGAGCTGCTCCCGTTCTGAGCGG |
| Sequence-based reagent | *trib1_fw* | This paper | PCR primer | TACGGGCATTTCACTTTCGG |
| Sequence-based reagent | *trib1_rev* | This paper | PCR primer | GTGAGGATCCCAGGAGACC |
| Sequence-based reagent | *trib3_fw* | This paper | PCR primer | ACCTGTTCAATCTTGTTGTCACA |

*Continued on next page*

*Continued*

| Reagent type (species) or resource | Designation | Source or reference | Identifiers | Additional information |
|---|---|---|---|---|
| Sequence-based reagent | *trib3_rev* | This paper | PCR primer | GGAAGGAGGCTGACTGAGTC |
| Sequence-based reagent | *cop1_fw* | This paper | PCR primer | TTCAATTATGTCAAGCACTCGG |
| Sequence-based reagent | *cop1_rev* | This paper | PCR primer | CAAGGGTCTTTTCCTGCTTAAA |
| Sequence-based reagent | *trib1cloning_fw* | This paper | PCR primer | TACGGGCATTTCACTTTCGG |
| Sequence-based reagent | *trib1cloning_rev* | This paper | PCR primer | CAGTCCTTAAACCCGACACG |
| Sequence-based reagent | *trib2cloning_fw* | This paper | PCR primer | CACCATGAACATACAGAGATCCAG |
| Sequence-based reagent | *trib2cloning_rev* | This paper | PCR primer | TTGCTACATCACTCAACGCC |
| Sequence-based reagent | *trib3cloning_fw* | This paper | PCR primer | CAACTAAGTGCGCCTGTAGT |
| Sequence-based reagent | *trib3cloning_rev* | This paper | PCR primer | TGCCCTTGAACTCTGCATAC |
| Genetic reagent (*Danio rerio*) | *tyrosinase* CRISPR-Cas9 guide RNA | **Isles et al., 2019** | *tyr* CRISPR | GGACTGGAGGACTTCTGGGG(AGG) |
| Strain, strain background (*Mycobacterium marinum*) | *Mycobacterium marinum* strain M (ATCC #BAA-535), containing the pSMT3-mCherry vector | **van der Sar et al., 2009** | *Mycobacterium marinum* strain M ATCC #BAA-535 | Transgenic |
| Antibody | Anti-nitrotyrosine antibody (rabbit polyclonal) | Merck | 06–284; Merck Millipore | (1:200) |
| Antibody | Anti-rabbit Alexa Fluor 633-conjugated secondary antibody (goat polyclonal) | | A-21071; ThermoFisher | (1:500) |
| Commercial assay, kit | DIG RNA Labeling Kit (SP6/T7) | Roche | 11175025910; Roche, Merck | |
| Commercial assay, kit | mMessageMachine SP6 Transcription Kit | Invitrogen | AM1340; Invitrogen | |
| Commercial assay, kit | QIAquick Gel Extraction Kit | QIAGEN | 28704; QIAGEN | |
| Commercial assay, kit | Zero Blunt TOPO PCR Cloning Kit | Invitrogen | K280002; Invitrogen | |
| Software, algorithm | BASiCz | Blood atlas of single cells in zebrafish | https://www.sanger.ac.uk/tool/basicz/ | |
| Software, algorithm | ChopChop | ChopChop | http://chopchop.cbu.uib.no/ | |
| Software, algorithm | Primer3 | ELIXIR | https://primer3.ut.ee/ | |
| Software, algorithm | LASX | Leica Microsystems | https://www.leica-microsystems.com/products/microscope-software/p/leica-las-x-ls/downloads/ | |
| Software, algorithm | AlphaFold DB | Deepmind/EMBL-EBI | https://alphafold.ebi.ac.uk/ | |
| Software, algorithm | ChimeraX | UCSF | https://www.cgl.ucsf.edu/chimerax/ | |
| Software, algorithm | Prism 10 | Graphpad | https://www.graphpad.com/ | |

## Materials availability statement

Further information and requests for resources and reagents developed here, including the newly created genetic tools for *tribbles* manipulation, should be directed to and will be fulfilled by the corresponding author Philip Elks (p.elks@sheffield.ac.uk).

## Human transcriptomic dataset analysis

Expression of TRIB1 in human CD14$^+$ monocytes and the site of a TST was derived from publicly available transcriptomic data deposited in EBI ArrayExpress repository (datasets E-MTAB-8162 and E-MTAB-6816, respectively – https://www.ebi.ac.uk/biostudies/arrayexpress) where the recruitment of patients with active and latent TB was approved by UK National Research Ethics Committees (reference numbers: 14/LO/0505, 16/LO/0776, and 18/LO/0680) and was subject to written informed consent (*Pollara et al., 2021*).

## Protein structure predictions

Predicated protein structures were sourced from AlphaFold DB (*Jumper et al., 2021*; *Varadi et al., 2022*). Overlays of predicted protein structures were produced using UCSF ChimeraX, developed by the Resource for Biocomputing, Visualization, and Informatics at the University of California, San Francisco, USA (*Pettersen et al., 2021*).

## Zebrafish

Zebrafish were raised in The Biological Services Aquarium (University of Sheffield, UK) and maintained according to standard protocols (https://zfin.org/) in Home Office approved facilities. All procedures were performed on embryos pre 5.2 dpf which were therefore outside of the Animals (Scientific Procedures) Act, to standards set by the UK Home Office. Emybros/larvae from clutches were randomly allocated into groups pre manipulation. Adult fish were maintained at 28°C with a 14/10 hr light/dark cycle. Nacre zebrafish were used as a wildtype. Transgenic zebrafish lines used in this study were *Tg(mpx:GFP)i114* (*Renshaw et al., 2006*), *TgBAC(il-1β:GFP)sh445* (*Ogryzko et al., 2019*), *Tg(mpeg:nlsclover)sh436* (*Bernut et al., 2019*), *Tg(mpeg1:mCherryCAAX)sh378* (*Bojarczuk et al., 2016*), *Tg(lyz:nfsB.mCherry)sh260* (*Buchan et al., 2019*), and *Tg(phd3:GFP)i144* (*Santhakumar et al., 2012*).

## CRISPR–Cas9 guide design and CRISPant generation

Transcript details for *trib1* (current Ensembl entry code is ENSDARG00000110963, but previously coded as ENSDARG00000076142 which is the identifier code used in RNAseq datasets), *trib2* (ENSDARG00000068179), and *trib3* (ENSDARG00000016200) were obtained from Ensembl genome browser (https://www.ensembl.org/). Only one transcript was identified per gene which was used for CRISPR–Cas9 guide design. The web tool ChopChop (https://chopchop.cbu.uib.no) was used to design guide RNAs and primers (*trib1*: guide RNA: AGCCCGTGAGCAGATGTCCGCGG, F primer: TACGGGCATTTCACTTTCGG, R primer: GTGAGGATCCCAGGAGACC, restriction digest: SacII. *trib3*: guide RNA: TCAACTCGCTTCAGTCGCAGTGG, F primer: ACCTGTTCAATCTTGTTGTCACA , R primer: GGAAGGAGGCTGACTGAGTC, restriction digest: Mwol. *cop1*: guide RNA: CGAGCTGCTCCCGTTCTGAGCGG, F primer: TTCAATTATGTCAAGCACTCGG , R primer: CAAGGGTCTTTTCCTGCTTAAA, restriction digest: Hyp188I).

To genotype first genomic DNA was extracted from 2 to 4 dpf larvae via incubation at 95°C in 100 µl of 50 mM NaOH for 20 min followed by the addition of 10 µl 1 M Tris–HCl (pH 8). PCR was then performed on genomic DNA with relevant primer pair and enzyme (NEB). Digests were run on a 2% (wt/vol) agarose gel (Sigma, Merck) at 100 V. Samples that were positive for CRISPR mutation were not digested by the restriction enzyme due to destruction of the restriction enzyme recognition site.

All guide RNAs (Sigma, Merck) were microinjected in the following injection mix: 1 µl 20 mM guide RNA, 1 µl 20 mM Tracr RNA (Sigma, Merck), 1 µl Cas9 (diluted 1:3 in diluent B, NEB), 1 µl water (water was replaced with 100 ng/µl *trib1* RNA for *cop1* experiments). A *tyrosinase* guide RNA (Sigma, Merck) control that has negligible effects on innate immunity was used as a negative CRISPR (*Isles et al., 2019*). Embryos were microinjected with 1 nl guide RNA mix at the single-cell stage to generate F0 CRISPants.

**Table 1.** Summary of CRISPR-Cas9 guideRNAs, relevant primers and restriction enzymes used for genotyping.

| Gene | guideRNA (5′–3′) | F primer (5′–3′) | R primer (5′–3′) | Enzyme |
|------|-------------------|-------------------|-------------------|--------|
| trib1 | AGCCCGTGAGCAGATGTCCGCGG | TACGGGCATTTCACTTTCGG | GTGAGGATCCCAGGAGACC | SacII |
| trib3 | TCAACTCGCTTCAGTCGCAGTGG | ACCTGTTCAATCTTGTTGTCACA | GGAAGGAGGCTGACTGAGTC | MwoI |
| cop1 | CGAGCTGCTCCCGTTCTGAGCGG | TTCAATTATGTCAAGCACTCGG | CAAGGGTCTTTTCCTGCTTAAA | Hyp188I |

Generation of trib1$^{-/-}$ mutant zebrafish trib1$^{-/-}$ (trib1$^{SH628}$/trib1$^{SH628}$) mutant embryos were generated by CRISPR–Cas9-mediated mutagenesis targeted around a SacII restriction site in the first exon of trib1 (using the method described by *Hruscha et al., 2013*) and the trib1 guide RNA sequence shown in *Table 1* and the Key resources table.

Injected F0 generation were raised to adulthood (~3 months until breeding age) before individual fish were outcrossed with a wildtype (Nacre) zebrafish line. A selection of the offspring (F1, 24 embryos) was genotyped at 2 dpf by extracting the pooled genomic DNA of three embryos with 8 replicates per pair mating. From the genomic DNA, PCR was performed using trib1 primer pair and an overnight restriction digest with SacII (NEB) at 37°C (Key resources table). Digests were run on a 2% (wt/vol) agarose gel at 100 V, positive mutations identified by undigested sample bands.

Batches of F1 embryos that were positive for mutations were pooled and raised. When raised F1 reached adulthood (~3 months), fish were fin clipped and genotyped. PCR was performed on the genomic DNA and purified using QIAquick PCR Purification Kit (QIAGEN) sequencing identified specific mutations in each founder. A male fish with a 14-bp deletion (loss of AGCAGATGTCCG CG) was outcrossed with female wildtype (Nacre) fish. Resulting F2 offspring were raised to adulthood and fin clipped and genotyped. Heterozygous trib1$^{+/-}$ mutant zebrafish (allele number sh628) were in-crossed to generate trib1$^{+/+}$, trib1$^{+/-}$, and trib$^{-/-}$ F3 sibling offspring that were used for Mm experiments.

## Cloning and whole-mount in situ hybridisation of trib 1, 2, and 3

RNA probes for zebrafish trib1 (ENSDARG00000110963), trib2 (ENSDARG00000068179), and trib3 (ENSDARG00000016200) were designed and synthesised after cloning the full-length genes into the pCRBlunt II-TOPO vector according to the instructions (ThermoFisher). Plasmid was linearised with the relevant restriction enzyme (trib1: HindIII, trib2: BsrGI, trib3: BsrGI) (NEB) and probes were synthesised according to DIG RNA Labelling Kit (SP6/T7, Roche, Merck). Zebrafish larvae were anaesthetised in 0.168 mg/ml Tricaine (MS-222, Sigma, Merck) in E3 media, which was removed and replaced with 4% (vol/vol in phosphate-buffered saline [PBS]) paraformaldehyde solution (Thermo Fisher) overnight at 4°C to fix. Whole-mount in situ hybridisation was performed as previously described (*Thisse and Thisse, 2008*).

## RNA injections for trib overexpression experiments

Forward inserts of trib1, trib2, and trib3 were cut from the pCRBlunt II-TOPO constructs using a double restriction digest with BamHI and XbaI at 37°C for 1.5 hr. The expression vector pCS2+ (Addgene) was digested using the same restriction enzyme pair and all digests were gel extracted using QIAquick Gel Extraction Kit (QIAGEN). Gel extracts of vector and trib digests were ligated into pCS2+ via overnight incubation at room temperature with T4 DNA ligase according to the manufacturer's instructions (NEB). Constructs were confirmed using sequencing performed by the University of Sheffield's Genomics core facility. RNA of each trib isoform was transcribed using mMessageMachine kit (Invitrogen) and diluted to 100 ng/µl in PR (diluted 1:10 in RNAse free water) for microinjection. Embryos were microinjected with 1 nl of 100 ng/µl RNA (measured using a 10-mm graticule) at the single-cell stage as previously described (*Elks et al., 2011*). RNA of dominant active (DA) and negative (DN) hif-1ab variants (ZFIN: hif1ab) were used for controls (*Elks et al., 2013*; *Elks et al., 2011*).

### *Mycobacterium marinum* culture and injection

Bacterial infection experiments were performed using *Mycobacterium marinum* strain M (ATCC #BAA-535), containing the pSMT3-mCherry vector (*van der Sar et al., 2009*). Liquid cultures were prepared

from bacterial plates before washing in PBS and diluting in 2% (wt/vol) polyvinylpyrrolidone40 (PVP40, Sigma, Merck) for injection as described previously (*Benard et al., 2012*). Injection inoculum was prepared to 100 colony-forming units (cfu)/nl for all burden experiments, loaded into borosilicate glass microcapillary injection needles (World Precision Instruments, pulled using a micropipette puller device, WPI) before microinjection into the circulation of 30 hpf zebrafish larvae via the caudal vein.

Prior to injection, zebrafish were anaesthetised in 0.168 mg/ml Tricaine (MS-222, Sigma, Merck) in E3 media and were transferred onto 1% agarose in E3 + methylene blue plates, removing excess media. All pathogens were injected using a microinjection rig (WPI) attached to a dissecting microscope. A 10-mm graticule was used to measure 1 nl droplets of injection volume, and for consistency, droplets were tested every 5–10 fish and the needle recalibrated if necessary. After injection, zebrafish were transferred to fresh E3 media for recovery and maintained at 28°C.

### Anti-nitrotyrosine immunostaining

Larvae were fixed in 4% (vol/vol) paraformaldehyde in PBS overnight at 4°C, and nitrotyrosine levels were immune labelled using immunostaining with a rabbit polyclonal anti-nitrotyrosine antibody (Merck) and detected using an Alexa Fluor-conjugated secondary antibody (Thermo Fisher) as previously described (*Elks et al., 2014*; *Elks et al., 2013*).

### Confocal microscopy

*TgBAC(il-1β:GFP)sh445* larvae and larvae immune-stained for nitrotyrosine were imaged using a Leica DMi8 SPE-TCS microscope using a HCX PL APO 40×/1.10 water immersion lens. Larvae were anaesthetised in 0.168 mg/ml Tricaine and mounted in 1% (wt/vol) low-melting agarose (Sigma, Merck) containing 0.168 mg/ml tricaine (Sigma, Merck) in 15μ-Slide 4 well glass bottom slides (Ibidi).

### Stereo microscopy

Zebrafish larvae were anaesthetised in 0.168 mg/ml Tricaine and transferred to a 50-mm glass bottomed FluoroDish (Ibidi). Zebrafish were imaged using a Leica DMi8 SPE-TCS microscope fitted with a Hamamatsu ORCA Flash 4.0 camera attachment using a HC FL PLAN 2.5×/0.07 dry lens. Whole-mount in situ staining was imaged using a Leica MZ10F stereo 14 microscope fitted with a GXCAM-U3 series 5MP camera (GT Vision).

### Image analysis

To calculate bacterial burden, fluorescent pixel count was measured using dedicated pixel count software (*Stoop et al., 2011*). For confocal imaging of anti-nitrotyrosine staining or transgenic lines, ImageJ (*Schindelin et al., 2012*) was used to quantify corrected total cell fluorescence (*Elks et al., 2014*; *Elks et al., 2013*).

### Statistical analysis

Statistical significance was calculated and determined using GraphPad Prism 10.0. Quantified data figures display all data points, with error bars depicting standard error of the mean unless stated otherwise in the figure legend. Statistical significance was determined using one-way analysis of variance with Bonferroni's multiple comparisons post hoc test/Kruskal–Wallis for experiments with three or more experimental groups, or paired/unpaired $T$-test/Wilcoxon matched pairs signed rank test for experiments with two experimental groups, unless stated otherwise in figure legend. p values shown are: *p < 0.05, **p < 0.01, and ***p < 0.001.

## Acknowledgements

The authors would like to thank Dr Heba Ismail, The University of Sheffield, for her expertise and helpful advice on E3 ubiquitin ligases. Thanks also to The Biological Services Aquarium Team at the University of Sheffield for their expert assistance with zebrafish husbandry. This work was supported by a University of Sheffield PhD scholarship awarded to FRH, PME, and AL are funded by a Sir Henry Dale Fellowship jointly funded by the Wellcome Trust and the Royal Society (Grant Number 105570/Z/14/Z/A). MN was funded by the Wellcome Trust (WT101766/Z/13/Z to GP and 207511/Z/17/Z to MN), Medical Research Council (MR_N007727_1 to GST), Academy of Medical Sciences (SGL021\1045) to GP, and National Institute for Health Research Biomedical Research Centre at University College London

Hospitals funding. Zebrafish infection work was performed in The Wolfson Laboratories for Zebrafish Models of Infection (The Wolfson Foundation/Royal Society grant number WLR\R1\170024) at the University of Sheffield.

## Additional information

### Funding

| Funder | Grant reference number | Author |
|---|---|---|
| Wellcome Trust | 10.35802/105570 | Philip M Elks |
| Wellcome Trust | 10.35802/101766 | Gabriele Pollara |
| Wellcome Trust | 10.35802/207511 | Mahdad Noursadeghi |
| Medical Research Foundation | MR_N007727_1 | Gillian S Tomlinson |
| Academy of Medical Sciences | SGL021\1045 | Gabriele Pollara |
| Wolfson Foundation | WLR\R1\170024 | Philip M Elks |

The funders had no role in study design, data collection, and interpretation, or the decision to submit the work for publication. For the purpose of Open Access, the authors have applied a CC BY public copyright license to any Author Accepted Manuscript version arising from this submission.

### Author contributions

Ffion R Hammond, Conceptualization, Resources, Data curation, Formal analysis, Validation, Investigation, Visualization, Methodology, Writing - original draft, Writing - review and editing; Amy Lewis, Conceptualization, Data curation, Formal analysis, Validation, Investigation, Visualization, Methodology, Writing - original draft, Project administration, Writing - review and editing; Gabriele Pollara, Conceptualization, Resources, Data curation, Formal analysis, Funding acquisition, Validation, Investigation, Visualization, Methodology, Writing - original draft, Writing - review and editing; Gillian S Tomlinson, Conceptualization, Resources, Data curation, Formal analysis, Funding acquisition, Validation, Investigation, Visualization, Methodology, Writing - review and editing; Mahdad Noursadeghi, Conceptualization, Resources, Data curation, Formal analysis, Funding acquisition, Validation, Investigation, Visualization, Methodology, Project administration, Writing - review and editing; Endre Kiss-Toth, Conceptualization, Data curation, Formal analysis, Supervision, Funding acquisition, Validation, Investigation, Visualization, Methodology, Writing - original draft, Project administration, Writing - review and editing; Philip M Elks, Conceptualization, Resources, Data curation, Formal analysis, Supervision, Funding acquisition, Validation, Investigation, Visualization, Methodology, Writing - original draft, Project administration, Writing - review and editing

### Author ORCIDs

Amy Lewis http://orcid.org/0000-0002-1156-3592
Gillian S Tomlinson http://orcid.org/0000-0003-4342-3161
Mahdad Noursadeghi http://orcid.org/0000-0002-4774-0853
Endre Kiss-Toth http://orcid.org/0000-0003-4406-4017
Philip M Elks http://orcid.org/0000-0003-1683-0749

### Ethics

Zebrafish were raised in The Biological Services Aquarium (University of Sheffield, UK) and maintained according to standard protocols (https://zfin.org/) in Home Office approved facilities. All procedures were performed on embryos pre 5.2 days post fertilisation (dpf) which were therefore outside of the Animals (Scientific Procedures) Act, to standards set by the UK Home Office. This work was performed under a UK Home Office Project Licence (PP7684817).

### Decision letter and Author response

Decision letter https://doi.org/10.7554/eLife.95980.sa1

Author response https://doi.org/10.7554/eLife.95980.sa2

## Additional files

### Supplementary files
• MDAR checklist

### Data availability
All data generated or analysed during this study are included in the manuscript, supporting files and source data. Expression data was derived from publicly available transcriptomic data deposited in EBI ArrayExpress repository (datasets E-MTAB-8162 and E-MTAB-6816 respectively - https://www.ebi.ac.uk/biostudies/arrayexpresshttps://www.ebi.ac.uk/biostudies/arrayexpress). Original gel images can be found in Figure 4—figure supplement 2—source data 1,2, Figure 4—figure supplement 3—source data 1 and Figure 8—figure supplement 1—source data 1. Figure 1—source data 1–9, Figure 4—source data 1–5, Figure 4—figure supplement 3—source data 2, Figure 5—source data 1–6, Figure 6—source data 1,2, Figure 8—source data 1,2, and Figure 8—figure supplement 1—source data 2 contain the numerical data used to generate the figures.

The following previously published datasets were used:

| Author(s) | Year | Dataset title | Dataset URL | Database and Identifier |
|---|---|---|---|---|
| Pollara G | 2021 | Monocytes from people with active or latent TB stimulated in vitro with PPD | https://www.ebi.ac.uk/biostudies/arrayexpress/studies/E-MTAB-8162 | EBI ArrayExpress, E-MTAB-8162 |
| Pollara G | 2019 | Transcriptional profiling of the tuberculin skin test in patients with active and latent tuberculosis | https://www.ebi.ac.uk/biostudies/arrayexpress/studies/E-MTAB-6816 | EBI ArrayExpress, E-MTAB-6816 |

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
