## [Editor Report]

This is a valuable study that defines the role of the pseudokinase Tribbles1 in the host response to mycobacteria. The data supports the function of Tribbles in a host protective, Cop-1 dependent, inflammatory response to Mycobacterium marinum in zebrafish and is deemed solid. This study would be of interest to scientists focused on host-pathogen interactions.

---

## [Decision Letter]

[Editors' note: this paper was reviewed by Review Commons.]

Thank you for resubmitting your work entitled "Tribbles1 is host protective during in vivo mycobacterial infection" for further consideration by *eLife*. Your revised article has been evaluated by Carla Rothlin (Senior Editor) and a Reviewing Editor.

The manuscript has been improved but there are some remaining issues that need to be addressed, as outlined below:

Reviewer comments:

The authors have addressed some of the points I made, but some points still need to be fully addressed.

1. The main article I was meaning with regard to Trib regulation was: Rishi, L., Hannon, M., Salomè, M., Hasemann, M., Frank, A.-K., Campos, J., Timoney, J., O'Connor, C., Cahill, M. R., Porse, B., and Keeshan, K. (2014) Regulation of Trib2 by an E2F1-C/EBPα feedback loop in AML cell proliferation, Blood 123, 2389-2400.

Admittedly I refer to human when the work is both mouse and human but the authors go into some depth about TRIB2 regulation/promoter regions. It would still be nice if the authors consider parallels between transcriptional regulation of human and zebrafish Tribbles, if they are aiming to promote a model system. There are also summaries of potential transcriptional sites in the different TRIB promoters in the articles below.

-Eyers, P. A., Keeshan, K., and Kannan, N. (2017) Tribbles in the 21st Century: The Evolving Roles of Tribbles Pseudokinases in Biology and Disease, Trends Cell Biol., Elsevier Ltd 27, 284-298.

-Salomé, M., Hopcroft, L., and Keeshan, K. (2018) Inverse and correlative relationships between TRIBBLES genes indicate non-redundant functions during normal and malignant hemopoiesis, Exp. Hematol. 66, 63-78.e13.

2. While the authors have added alphafold models they are not presented in a manner that builds confidence that the overlays are performed correctly. The same orientation should be used for each of the panels E-G, and the overlays look surprisingly poor given the sequence homology. It looks to me that the authors are trying to superimpose the entire structure of TRIB2, when they should superimpose just the C-terminal domains. I would strongly suggest the authors consult a local structural biologist in making sure this is right.

---

## [Author Response]

1. General Statements

In this paper we report the discovery that a member of the tribbles pseudokinase family, TRIB1 is expressed in human monocytes and is upregulated after stimulation with mycobacterial antigen in a human patient challenge model, the first direct link between immune cell Tribbles expression and innate immune response to infection. We then interrogated the mechanisms of Tribbles roles in TB using a human disease relevant whole-organism in vivo zebrafish model of TB. We show that specifically TRIB1 modulation can tip the battle between host and pathogen enhancing the innate immune response and reducing bacterial burden. We then uncover the *molecular mechanisms* responsible for the host protective effect of TRIB1, with enhanced antimicrobial reactive nitrogen species and il-1beta, via cooperation with Cop1 E3 ubiquitin ligase. Our findings demonstrate, for the first time, TRIB1 as a host moderator of antimicrobial mechanisms, whose manipulation is of benefit to the host during mycobacterial infection and as such, a potential novel therapeutic target against TB infection.

We thank the reviewers for their positive appraisal of our work and for their helpful suggestions that will improve our manuscript. In particular we would like to highlight the reviewer’s comments on the gap/need for a new zebrafish in vivo model to understand the roles of tribbles in infection that can “be extrapolated into the human system”, and how they feel these findings will be of broad interest and “significance to cross section of the research community” attracting “interest from readers in the fields of infection, immunity, hematology and animal models” alongside “researchers studying all aspects of Tribbles pseudokinase function, especially researchers seeking models to test small molecule agonists and antagonists.”

2. Point-by-point description of the revisionsReviewer 1Major commentsThe major weakness of the manuscript is that the authors do not evaluate C/EBP transcription factors at all. It is rather surprising as they emphasize cooperation between Trib1 and Cop1 in the main title. C/EBP family proteins are key factors of Trib1-mediated modulation of granulocytes and monocytes. Also, slbo, a *Drosophila* homolog of C/EBP, is a target of tribbles, indicating that the pathway is evolutionary conserved. I would request the following experiments and discussions.1. The authors should investigate the expression of the C/EBPa protein p42 isoform and/or other C/EBP family proteins such as C/EBPb, and confirm that the p42 is degraded by Trib1 overexpression and recovered by Trib1 and Cop1 knockout. It is also important to determine both p42 and p30 isoforms are preserved in zebrafish.

We agree that possible C/EBP roles should be discussed in detail and we have added to a reworked Discussion section between lines 407-421.

We stand by our data that the host protective mechanism of Trib1 acts requires Cop1, but we are not able to directly show a C/EBP mechanism within the scope of the current project due to a lack of tools/knowledge in the zebrafish on this (as outlined in our Review Commons Revision Plan). It is important to note that we have not claimed a C/EBP mechanism in our manuscript, and we think it is possibly unlikely given that monocyte and granulocyte numbers are not altered after TRIB1 manipulation (including new data outlined in comments below). There are many other candidates other than C/EBP that COP1 could be acting through. Some examples include MAPK (Niespolo et al., Front Immunol, 2020), JAK/Stat (Arndt er al., 2018), serine threonine kinases (Durzynska et al., Structure, 2017) and β-Catenin (Zahid et al., Proteins, 2022), pathways that are challenging to dissect in an in vivo setting. There is also evidence suggesting that COP1 and C/EBP have distinct binding sites on TRIB1, potentially uncoupling their activity in some biological situations (Murphy et al., Structure, 2015). We have now fully discussed this between lines 407-421.

We do not have antibodies or tools to detect p42 and p30 in zebrafish. As Tribbles1 regulation of C/EBPa appears to be post-translational (Bauer et al., J Clin Invest, 2015), this would be incredibly challenging to unpick in the zebrafish model due to lack of tools to do this.

In response to this comment, we have modified the title from “Tribbles1 and Cop1 cooperate to protect the host during in vivo mycobacterial infection” to “Tribbles1 is host protective during in vivo mycobacterial infection”. We believe our data does show that the protective effect of Tribbles requires Cop1, but changing the title in this way removes any suggestion that they directly cooperate in a potential C/EBP dependent manner, suggested by the reviewer. We have also further addressed this experimentally in the following reviewer comment, below.

2. Although the authors found the number of neutrophils and monocytes unchanged by Trib1 overexpression nor knockdown, they did not demonstrate the differentiation status of both cell types. This is quite an important issue, given that Trib1 knockout promotes granulocytic differentiation via C/EBPa accumulation in mice. Also, the analysis of granulocytic/monocytic differentiation will provide the crucial information how Trib1 protects the host from mycobacterial infection regulating hematopoietic cell functions. The authors should perform morphological analysis and examine cell surface marker expression to examine whether Trib1 and Cop1 modulates granulocytic and monocytic differentiation with and without Mm infection.

Unfortunately, we do not have the same level of immunology knowledge nor the antibodies to look at cell surface markers in zebrafish larvae (it is noted that the reviewer identifies that they “not have sufficient expertise in zebrafish models.” We agree with the reviewer that this would be an obvious and informative experiment to do in mouse models, but is not currently possible in zebrafish larval models). The transgenic promoters used (mpx for neutrophils and mpeg1 for macrophages) are robust and widely published to look at total neutrophil and macrophage numbers (Renshaw et al., Blood 2006; Ellett et al., Blood 2011). Mpx, encoding myeloperoxidase, is expressed late in neutrophil differentiation.

In response to this comment, we have performed new experiments in a number of areas:

1. We have generated a new stable Tribbles 1CRISPR-Cas9 knockout mutant and assessed neutrophil differentiation using Sudan Black (SD). SD stains neutrophil granules, the development of which is during a late phase of neutrophil differentiation. In new Figure 4—figure supplement 3F-G we show that in the stable *trib1-/-* mutant there is no difference in the neutrophil number at 2dpf.

2. We have investigated modulation of tribbles at the later timepoint of 3dpf to add to our 2dpf data, and show that no difference in the number of *mpx:GFP* neutrophils after either *trib1* overexpression (new Figure 5E) or *trib1* knockdown (new Figure 5F) at this later timepoint when alterations in myelopoiesis would be more obvious if present.

3. Interestingly, it has been shown that a zebrafish myeloid expressed C/EBP, c/ebp1, is not required for initial macrophage or granulocyte development, but its knockdown did result in reduced expression of the secondary granule gene LysC, shown by in situ hybridisation (Su et al., Zebrafish, 2007). To test this further we used an *LysC:mCherry* transgenic line (Buchan et al., PLoS One 2019) to assess expression in neutrophils after *trib1* manipulation and found no difference in the number of cells expressing LysC after either *trib1* overexpression (new Figure 5G) or *trib1* knockdown (new Figure 5H) at either 2 or 3dpf. This is consistent with *trib1* not acting via c/ebp1 in our models, at least at these stages of development.

We cannot perform accurate leukocyte counts during Mm infection reliably as neutrophils/macrophages cluster around infected areas and undergo cell death processes making counting challenging.

C/EBPa is found in zebrafish and is involved in myeloid differentiation and haematopoeisis (Yuan et al., Blood 2011). The most involved C/EBP in zebrafish myelopoiesis is a zebrafish specific isoform called c/ebp1 that is myeloid expressed (Lyons et al., Blood 2001). This has a highly conserved carboxy-yterminal bZIP domain but the amino-terminal domains are unique. A new manuscript has demonstrated that c/ebp1 controls eosinophilopoiesis in zebrafish (Li et al., Nature Communications, 2024 https://www.nature.com/articles/s41467-024-45029-0), and also neutrophil development, but that this only occurs from 5dpf onwards, therefore outside of the time window we are investigating in our study. This study, alongside the additional data we have collected, suggests that *trib1* is not acting via c/ebp1 to affect neutrophil differentiation at the timepoints investigated in our study. We have added these observations in the Results section and in the reworked discussion.

3. It is interesting that Cop1 knockdown zebrafish is viable, given its ubiquitous expression and multiple important targets of protein degradation. The authors should provide the details of phenotype of Cop1 KO larva and discuss on this issue.

Zebrafish mutants are much less prone to embryonic lethality compared to mice as maternally contributed protein stores allow for basic metabolic functions to occur throughout the short period of embryonic development and their development is completely external without the need for a placenta and associated gene expression (Rossant and Hopkins. Genes and Development 1992). However, in the case of the *cop1* CRISPant, this is a knockdown rather than a knockout, so there may be sufficient remaining Cop1 availability for development if it is indeed a requirement for larval viability. Although Cop1 knockout mice are non-viable, hypomorphs (similar to our knockdown zebrafish) are viable and develop relatively normally but are tumour prone as Cop1 is required for effective tumour suppression (Milgliorini et al., JCI, 2011).

We had not commented on the Cop1 larvae phenotype as they look like they develop normally eg. normal body axis, development. We now briefly comment on this in the Results section on line 306. Furthermore, we have now added wholebody neutrophil counts into new Figure 8—figure supplement 1C and show there is no change with *cop1* knockdown, suggesting no difference in granulopoiesis/neutrophil development.

4. [Optional] The effect of enhanced ERK phosphorylation by Trib1 for the protective effect against mycobacterial infection is another interesting point. It would be better if the authors could provide the ERK phosphorylation status upon Trib1 overexpression.

Unfortunately, we have no method to answer this question conclusively within the scope of this project (as outlined in our Review Commons Revision Plan). There are limited reports of phosphorylated ERK antibodies that work in wholemount zebrafish (eg, Maurer and Sagerström, BMC Developmental Biology, 2018, that use a rabbit antibody), but this is widely expressed in many tissues of the zebrafish and immune cells would be challenging to resolve.

5. [Optional] To obtain the more solid evidence for the Cop1 dependent function of Trib1 on mycobacteria infection, it is better to use the Trib1 mutant that loses the Cop1 binding activity. This experiment will strength the authors' conclusion of the Trib1 and Cop1 cooperation.

We have addressed this comment by generating a new stable zebrafish CRISPR-Cas9 Tribbles 1 knockout line with a 14 base pair deletion that is predicted to lead a premature stop at 94aa in the middle of the pseudokinase domain, lacking the catalytic loop (new Figure 4—figure supplement 3 A-D). The predicted truncated protein lacks the predicted COP1 binding area at the C terminal of the protein. We found that the stable mutant also leads to an increase in bacterial burden compared to wildtype sibling controls (new Figure 4E). As discussed above, we use Sudan Black staining in the stable mutant to demonstrate no change in neutrophil numbers (new Figure 4—figure supplement 3F-G).

Reviewer #2The paper by Hammond et al. characterises the role of Tribbles proteins in Zebrafish in response to MTb infection. They show interesting upregulation of TRIB1 and 2, relative to TRIB3 which is not upregulated. This may provide an interesting system to further explore the role of Tribbles in response to infection, which is currently underexplored, but could do with some additional detail to stake that claim more strongly.Major CommentsSome discussion of the mechanisms regulating TRIB1/2/3 transcriptionally is probably relevant given the differential upregulation observed during infection. There is quite a bit of characterisation of different Tribble promoter regions in humans-how does this translate to Zebrafish?

We are not sure exactly which studies characterising tribbles promotors the reviewer was referring to here. In response to this comment we interrogated a variable number tandem repeat identified in human tribbles 3 (Örd et al., 2020) and searched for this in the first 50,000 base pairs of the zebrafish *tribbles 3* promotor but did not identify this mammalian-conserved sequence. We added this detail in the Results section on lines 160-165. Further specific sequence details on other promotor region characterisation of Tribbles in the literature was not obvious to us for us to perform further searches.

Structural comparisons are relatively descriptive of identity etc. Nowadays it should be relatively straightforward to comment on structural conservation based on Alphafold models. Specific details may not be accurate but gross folds will be, and comparing those may be more informative.

We thank the reviewer for this suggestion as it has added useful information to our manuscript. We have now added overlays of Alphafold predictions of zebrafish Trib1, 2 and 3, with human TRIB 1, 2 and 3 in new Figure 2E-G. Gross folds and α chains are conserved between tribbles 1 and 3 zebrafish and human predications. Tribbles 2 orthologues did not overlay well, but this is most likely due to the shorter nature of the zebrafish Trib2 amino acid sequence (343 in human compared to 207 in zebrafish).

In terms of Crispr use-can it be confirmed that Crispr modified cell lines have effects at the protein level? This is not my specific expertise, but the supplementary evidence shown seems to show some genomic editing is occurring, but not necessairly how it effects protein levels.

We do not have antibodies that work on zebrafish Tribbles proteins to assess this directly at the protein level. However, we have now addressed this comment by generating a new stable zebrafish CRISPR Tribbles 1 knockout line to support our knockdown studies, with a 14 base pair deletion that is predicted to lead a premature stop at 94aa in the middle of the pseudokinase domain, lacking the catalytic loop (new Figure 4—figure supplement 3 A-D). Unlike the “CRISPant” knockdown work in the peer-reviewed version, this represents a stable knockout of Tribbles 1. We show in new Figure 4E that this full knockout has a similar effect on bacterial burden to the knockdown corroborating our findings in the CRISPant knockdown.

While the protective effect is stated as an effect size 'close to that of HIF-1a', is there additional rationale suggesting that the two may be linked?

Yes, there have been a number of studies that link Tribbles and Hif1-α. The best characterised link is in different cancer cells where Tribbles 3 has been linked to HIF-1alpha or hypoxia in breast cancer (Wennemers, Breast Cancer Research 2011), renal cell carcinoma cells (Hong et al., Inj J Biol Sci, 2019) and adenocarcinoma (Xing et al., Cancer Management Research, 2020). In *Drosophila* Hif-1alpha induces TRIB in fat body tissue (Noguchi et al., Genes Cells 2022). We have now added references to these studies to the relevant section in the results (lines 287-290).

Reviewer #3Summary:In this manuscript, the authors test the role of Tribbles psuedokinase 1 in the primary immune defence against Mycobacterium tuberculosis, the pathogen in tuberculosis. After showing increased Trib1 and 2 in response to mycobacterial infection cell culture and from biopsies of challenged human tissue, they turn to a zebrafish model of infection of the caudal vein with Mycobacterium marinum, a natural fish pathogen similar in effects on macrophages to M. tuberculosis in humans.Authors find that overexpression of tribs 1, 2 and 3 had no strong effect on development but reduced the bacterial burder significantly for Trib1, less so for Trib2 and not at all for Trib3. Conversely, Trib1 Crisper-mediated knockdown increased Mm burde, which 3 had no effect ( and 2 guide RNAs were not effective at reducing Trib2).Trib1 increased levels of pro-inflammatory interleukin 1 β, as measure by a reporter gne , which Trib3 had no similar effect, which was on par to the effect of the Hif-1alpha transcription factor, known to regulate il-1beta. The effect of Trib1 was not upon increased Hif-a (hence independent) but was dependent on the co-factor COP1, a known target of the Trib1 C-tail when activatedNo major problems seen

[Editors’ note: what follows is the authors’ response to the second round of review.]

The manuscript has been improved but there are some remaining issues that need to be addressed, as outlined below:Reviewer comments:The authors have addressed some of the points I made, but some points still need to be fully addressed.1. The main article I was meaning with regard to Trib regulation was: Rishi, L., Hannon, M., Salomè, M., Hasemann, M., Frank, A.-K., Campos, J., Timoney, J., O'Connor, C., Cahill, M. R., Porse, B., and Keeshan, K. (2014) Regulation of Trib2 by an E2F1-C/EBPα feedback loop in AML cell proliferation, Blood 123, 2389-2400.Admittedly I refer to human when the work is both mouse and human but the authors go into some depth about TRIB2 regulation/promoter regions. It would still be nice if the authors consider parallels between transcriptional regulation of human and zebrafish Tribbles, if they are aiming to promote a model system. There are also summaries of potential transcriptional sites in the different TRIB promoters in the articles below.-Eyers, P. A., Keeshan, K., and Kannan, N. (2017) Tribbles in the 21st Century: The Evolving Roles of Tribbles Pseudokinases in Biology and Disease, Trends Cell Biol., Elsevier Ltd 27, 284-298.-Salomé, M., Hopcroft, L., and Keeshan, K. (2018) Inverse and correlative relationships between TRIBBLES genes indicate non-redundant functions during normal and malignant hemopoiesis, Exp. Hematol. 66, 63-78.e13.

We think this is an interesting topic, but we believe that to produce any meaningful insights on this is out of the scope of this manuscript as not enough is known about transcriptional regulation in zebrafish to directly compare it to human and mouse. We searched for any data (eg ChIP-seq) in the literature for zebrafish on the E2F and C/EBP binding sites focussed on in Rishi et al., Blood 2014 in zebrafish tribbles promoters and were unable to find any data. We also aimed to recapitulate the analysis of the human TRIB2 promoter outlined in Eyers et al., Trends Cell Biol 2017, to find putative transcriptional sites in the zebrafish *trib* promoters, but hit a number of stumbling blocks. The Transcription Element Search System (TESS) software used in Eyers et al., 2017 is no longer available and so we turned to other software. When searching for zebrafish transcriptional binding sites using PROMO software (https://alggen.lsi.upc.es/cgi-bin/promo_v3/; Martinez et al., Bioinformatics, 2002) we found no sites in 5kb of the promotors of zebrafish tribbles genes. We believe this to be because of the paucity of known transcriptional binding consensus sequences in the zebrafish. Indeed it was a similar story in JASPAR2024 (Rauluseviciuite et al., Nucleic Acids Research, 2024) where there are no zebrafish transcriptional binding sites in their “core” nor “invalidated” profiles databases. We also tried TFBind (Tsunoda and Takagi, Bioinformatics 1999) and Tfsitescan (http://www.ifti.org/cgi-bin/ifti/Tfsitescan.pl), which again do not have zebrafish consensus binding sites.

As none of our major findings are on TRIB2 (where the main data for the transcription site analysis exists in human/mice) we do not believe this to impact on our findings or conclusions and do not want to speculate in this area when so little is known in the zebrafish model.

2. While the authors have added alphafold models they are not presented in a manner that builds confidence that the overlays are performed correctly. The same orientation should be used for each of the panels E-G, and the overlays look surprisingly poor given the sequence homology. It looks to me that the authors are trying to superimpose the entire structure of TRIB2, when they should superimpose just the C-terminal domains. I would strongly suggest the authors consult a local structural biologist in making sure this is right.

Apologies, we have made an error here, thank you for spotting this. We did consult a structural biologist (as correctly inferred, this is not our area of expertise) and on their advice we used the widely-published ChimeraX, open source software to make the overlays. However, I am embarrassed to admit, we had missed an important step in the overlay process and, after further consultation, have now corrected this error in the newly revised Figure 2 (which also corrects for the overlay of TRIB2 only overlaying the domains which are present in both zebrafish and human). We have also changed the orientation of the overlays in the figures to the orientation that is standard in the literature with the N-lobe at the top and the C-lobe at the bottom (eg, McMillan et al., Cancers 2021; Murphy et al., Structure 2015; Eyers et al., Trends in Cell Biology 2017). Please note that we have provided the.cxs files in the source data for readers to observe the overlays in 3D using freely available software, so these overlays are available to readers in all orientations.